# ON THE IMPACT OF CLIENT SAMPLING ON FEDERATED LEARNING CONVERGENCE

## ABSTRACT

While clients' sampling is a central operation of current state-of-the-art federated learning (FL) approaches, the impact of this procedure on the convergence and speed of FL remains under-investigated. In this work we introduce a novel decomposition theorem for the convergence of FL, allowing to clearly quantify the impact of client sampling on the global model update. Contrarily to previous convergence analyses, our theorem provides the exact decomposition of a given convergence step, thus enabling accurate considerations about the role of client sampling and heterogeneity. First, we provide a theoretical ground for previously reported experimental results on the relationship between FL convergence and the variance of the aggregation weights. Second, we prove for the first time that the quality of FL convergence is also impacted by the resulting *covariance* between aggregation weights. Our theory is general, and is here applied to Multinomial Distribution (MD) and Uniform sampling, the two default client sampling schemes of FL, and demonstrated through a series of experiments in non-iid and unbalanced scenarios. Our results suggest that MD sampling should be used as default sampling scheme, due to the resilience to the changes in data ratio during the learning process, while Uniform sampling is superior only in the special case when clients have the same amount of data.

## 1 INTRODUCTION

Federated Learning (FL) has gained popularity in the last years as it enables different clients to jointly learn a global model without sharing their respective data. Among the different FL approaches, federated averaging (FedAvg) has emerged as the most popular optimization scheme (McMahan et al., 2017). An optimization round of FedAvg requires data owners, also called clients, to receive from the server the current global model which they update on a fixed amount of Stochastic Gradient Descent (SGD) steps before sending it back to the server. The new global model is then created as the weighted average of the client updates, according to their data ratio.

FedAvg was first proven to converge experimentally (McMahan et al., 2017), before theoretical guarantees were provided for any non-iid federated dataset (Wang et al., 2020a; Karimireddy et al., 2020; Haddadpour & Mahdavi, 2019; Khaled et al., 2020). A drawback of naive implementations of FedAvg consists in requiring the participation of all the clients at every optimization round. As a consequence, the efficiency of the optimization is limited by the communication speed of the slowest client, as well as by the server communication capabilities. To mitigate this issue, the original FedAvg algorithm already contemplated the possibility of considering a random subset of $m$ clients at each FL round. It has been subsequently shown that, to ensure the convergence of FL to its optimum, clients must be sampled such that in expectation the resulting global model is identical to the one obtained when considering all the clients (Wang et al., 2020a; Cho et al., 2020). Clients sampling schemes compliant with this requirement are thus called *unbiased*. Due to its simplicity and flexibility, the current default unbiased sampling scheme consists in sampling $m$ clients according to a Multinomial Distribution (MD), where the sampling probability depends on the respective data ratio (Li et al., 2020a; Wang et al., 2020a; Li et al., 2020c; Haddadpour & Mahdavi, 2019; Li et al., 2020b; Wang & Joshi, 2018; Fraboni et al., 2021). Nevertheless, when clients have identical amount of data, clients can also be sampled uniformly without replacement (Li et al., 2020c; Karimireddy et al., 2020; Reddi et al., 2021; Rizk et al., 2020). In this case, Uniform sampling has been experimentally shown to yield better results than MD sampling (Li et al., 2020c).

In spite of the practical usage of sampling in FL, the impact of different unbiased client sampling schemes on the convergence and speed of FL remains to date under-investigated. In (Fraboni et al., 2021), MD sampling was extended to account for collections of sampling distributions with varying client sampling probability, to define clustered sampling. From a theoretical perspective, this approach was proven to have identical convergence guarantees of MD sampling, with albeit experimental improvement justified by lower variance of the clients' aggregation weights. Another study investigated the convergence guarantees of FedAvg under different sampling schemes, through a penalized optimization framework (Li et al., 2020c). These studies reflect the ongoing interest and need for a theoretical framework to elucidate the impact of client sampling on FL convergence.

The main contribution of this work consists in deriving a decomposition theorem for the convergence of FL, allowing to clearly quantify the impact of client sampling on the global model update at any FL round. This contribution has important theoretical and practical implications. First, we demonstrate the dependence of FL convergence on the variance of the aggregation weights. Second, we prove for the first time that the convergence speed is also impacted through sampling by the resulting *covariance* between aggregation weights. Contrarily to the convergence bound illustrated in (Li et al., 2020c), our theorem provides the exact decomposition of a given convergence step, thus enabling accurate quantification of the role of client sampling and heterogeneity in FL. From a practical point of view, we establish both theoretically and experimentally that client sampling schemes based on aggregation weights with sum different than 1 are less efficient. We also prove that MD sampling is outperformed by Uniform sampling only when clients have identical data ratio. Finally, we show that the comparison between different client sampling schemes is appropriate only when considering a small number of clients. Our theory ultimately shows that MD sampling should be used as default sampling scheme, due to the favorable statistical properties and to the resilience to FL applications with varying data ratio and heterogeneity.

Our work is structured as follows. In Section 2, we provide formal definitions for FL, unbiased client sampling, and for the server aggregation scheme. In Section 3, we introduce our decomposition theorem (Theorem 1) relating the convergence of FL to the aggregation weight variance of the client sampling scheme. Consistently with our theory, in Section 4, we experimentally demonstrate the importance of the clients aggregation weights variance and covariance on the convergence speed, and conclude by recommending Uniform sampling for FL applications with identical client ratio, and MD sampling otherwise.

## 2 BACKGROUND

Before investigating in Section 3 the impact of client sampling on FL convergence, we recapitulate in Section 2 the current theory behind FL aggregation schemes for clients local updates. We then introduce a formalization for *unbiased* client sampling.

### 2.1 AGGREGATING CLIENTS LOCAL UPDATES

In FL, we consider a set $I$ of $n$ clients each respectively owning a dataset $\mathcal{D}_i$ composed of $n_i$ samples. FL aims at optimizing the average of each clients local loss function weighted by $p_i$ such that $\sum_{i=1}^{n} p_i = 1$, i.e.

$$\mathcal{L}(\theta) = \sum_{i=1}^{n} p_i \mathcal{L}_i(\theta), \tag{1}$$

where $\theta$ represents the model parameters. The weight $p_i$ can be interpreted as the importance given by the server to client $i$ in the federated optimization problem. While any combination of $\{p_i\}$ is possible, we note that in practice, either (a) every device has equal importance, i.e. $p_i = 1/n$, or (b) every data point is equally important, i.e. $p_i = n_i/M$ with $M = \sum_{i=1}^{n} n_i$. Unless stated otherwise, in the rest of this work, we consider to be in case (b), i.e. $\exists i, \; p_i \neq 1/n$.

In this setting, to estimate a global model across clients, FedAvg (McMahan et al., 2017) is an iterative training strategy based on the aggregation of local model parameters. At each iteration step $t$, the server sends the current global model parameters $\theta^t$ to the clients. Each client updates the respective model by minimizing the local cost function $\mathcal{L}_i(\theta)$ through a fixed amount $K$ of SGD steps initialized with $\theta^t$. Subsequently each client returns the updated local parameters $\theta_i^{t+1}$ to the

server. The global model parameters $\theta^{t+1}$ at the iteration step $t+1$ are then estimated as a weighted average:

$$\theta^{t+1} = \sum_{i=1}^{n} p_i \theta_i^{t+1}. \tag{2}$$

To alleviate the clients workload and reduce the amount of overall communications, the server often considers $m \leq n$ clients at every iteration. In heterogeneous datasets containing many workers, the percentage of sampled clients $m/n$ can be small, and thus induce important variability in the new global model, as each FL optimization step necessarily leads to an improvement on the $m$ sampled clients to the detriment of the non-sampled ones. To solve this issue, Reddi et al. (2021); Karimireddy et al. (2020); Wang et al. (2020b) propose considering an additional learning rate $\eta_g$ to better account for the clients update at a given iteration. We denote by $\omega_i(S_t)$ the stochastic aggregation weight of client $i$ given the subset of sampled clients $S_t$ at iteration $t$ . The server aggregation scheme can be written as:

$$\theta^{t+1} = \theta^t + \eta_g \sum_{i=1}^{n} \omega_i(S_t)(\theta_i^{t+1} - \theta^t). \tag{3}$$

## 2.2 UNBIASED CLIENTS SAMPLING

While FedAvg was originally based on the uniform sampling of clients (McMahan et al., 2017), this scheme has been proven to be biased and converge to a suboptimal minima of problem (1) (Wang et al., 2020a; Cho et al., 2020; Li et al., 2020c). This was the motivation for Li et al. (2020c) to introduce the notion of *unbiasedness*, where clients are considered in expectation subject to their importance $p_i$, according to Definition 1 below. Unbiased sampling guarantees the optimization of the original FL cost function, while minimizing the number of active clients per FL round. We note that unbiased sampling is not necessarily related to the clients distribution, as this would require to know beforehand the specificity of the clients' datasets.

Unbiased sampling methods (Li et al., 2020a;c; Fraboni et al., 2021) are currently among the standard approaches to FL, as opposed to *biased* approaches, known to over- or under-represent clients and lead to suboptimal convergence properties (McMahan et al., 2017; Nishio & Yonetani, 2019; Jeon et al., 2020; Cho et al., 2020), or to methods requiring additional computation work from clients (Chen et al., 2020a).

**Definition 1** (Unbiased Sampling). *A client sampling scheme is said unbiased if the expected value of the client aggregation is equal to the global deterministic aggregation obtained when considering all the clients, i.e.*

$$\mathbb{E}_{S_t} \left[ \sum_{i=1}^{n} w_i(S_t)\theta_i^t \right] := \sum_{i=1}^{n} p_i \theta_i^t, \tag{4}$$

*where $w_j(S_t)$ is the aggregation weight of client $j$ for subset of clients $S_t$.*

The sampling distribution uniquely defines the statistical properties of stochastic weights. In this setting, unbiased sampling guarantees the equivalence between deterministic and stochastic weights in expectation. Unbiased schemes of primary importance in FL are MD and Uniform sampling, for which we can derive a close form formula for the aggregation weights :

**MD sampling**. This scheme considers $l_1, ..., l_m$ to be the $m$ iid sampled clients from a Multinomial Distribution with support on $\{1, ..., m\}$ satisfying $\mathbb{P}(l_k = i) = p_i$ (Wang et al., 2020a; Li et al., 2020a;c; Haddadpour & Mahdavi, 2019; Li et al., 2020b; Wang & Joshi, 2018; Fraboni et al., 2021). By definition, we have $\sum_{i=1}^{n} p_i = 1$, and the clients aggregation weights take the form:

$$\omega_i(S_t) = \frac{1}{m} \sum_{k=1}^{m} \mathbb{I}(l_k = i). \tag{5}$$

**Uniform sampling**. This scheme samples $m$ clients uniformly without replacement. Since in this case a client is sampled with probability $p(\{i \in S_t\}) = m/n$, the requirement of Definition 1 implies:

$$\omega_i(S_t) = \mathbb{I}(i \in S_t)\frac{n}{m}p_i. \tag{6}$$

Table 1: Synthesis of statistical properties of different sampling schemes.

| Sampling | $\text{Var}\left[\omega_i(S_t)\right]$ | $\alpha$ | $\text{Var}\left[\sum_{i=1}^{n}\omega_i(S_t)\right]$ |
|---|---|---|---|
| Full participation | $= 0$ | $= 0$ | $= 0$ |
| MD | $= -\frac{1}{m}p_i^2 + \frac{1}{m}p_i$ | $= 1/m$ | $= 0$ |
| Uniform | $= \left(\frac{n}{m} - 1\right)p_i^2$ | $= \frac{n-m}{m(n-1)}$ | $= \frac{n-m}{m(n-1)}[n\sum_{i=1}^{n}p_i^2 - 1]$ |

We note that this formulation for Uniform sampling is a generalization of the scheme previously used for FL applications with identical client importance, i.e. $p_i = 1/n$ (Karimireddy et al., 2020; Li et al., 2020c; Reddi et al., 2021; Rizk et al., 2020). We note that $\text{Var}\left[\sum_{i=1}^{n}\omega_i(S_t)\right] = 0$ if and only if $p_i = 1/n$ for all the clients as, indeed, $\sum_{i=1}^{n}\omega_i(S_t) = m\frac{n}{m}\frac{1}{n} = 1$

With reference to equation (3), we note that by setting $\eta_g = 1$, and by imposing the condition $\forall S_t$, $\sum_{i=1}^{n}\omega_i(S_t) = 1$, we retrieve equation (2). This condition is satisfied for example by MD sampling and Uniform sampling for identical clients importance.

We finally note that the covariance of the aggregation weights for both MD and Uniform sampling is such that there exists an $\alpha$ such that $\forall i \neq j$, $\text{Cov}\left[\omega_i(S_t), \omega_j(S_t)\right] = -\alpha p_i p_j$. We provide in Table 1 the derivation of $\alpha$ and the resulting covariance for these two schemes with calculus details in Appendix A. Furthermore, this property is common to a variety of sampling schemes, for example based on Binomial or Poisson Binomial distributions (detailed derivations can be found in Appendix A). Following this consideration, in addition to Definition 1, in the rest of this work we assume the additional requirement for a client sampling scheme to satisfy $\exists \alpha \geq 0$, $\forall i \neq j$, $\text{Cov}\left[\omega_i(S_t), \omega_j(S_t)\right] = -\alpha p_i p_j$.

## 3 CONVERGENCE GUARANTEES

Based on the assumptions introduced in Section 2, in what follows we elaborate a new theory relating the convergence of FL to the statistical properties of client sampling schemes. In particular, Decomposition Theorem 1 describes the impact of client sampling on a single optimization step, while Theorem 2 quantifies the asymptotic relationship between client sampling and FL convergence.

### 3.1 DECOMPOSITION THEOREM

**Assumption 1** (Unbiased Gradient and Bounded Variance). *Every client stochastic gradient $g_i(\boldsymbol{x}|B)$ of a model $\boldsymbol{x}$ evaluated on batch $B$ is an unbiased estimator of the local gradient. We thus have $\mathbb{E}_B\left[\boldsymbol{\xi}_i(B)\right] = 0$ and $0 \leq \mathbb{E}_B\left[\|\boldsymbol{\xi}_i(B)\|^2\right] \leq \sigma^2$, with $\boldsymbol{\xi}_i(B) = g_i(\boldsymbol{x}|B) - \nabla\mathcal{L}_i(\boldsymbol{x})$.*

the following Decomposition Theorem highlights the impact of client sampling on a single FL optimization step. For any optimization round, it provides an explicit link between the statistical properties of the sampling scheme (variance and covariance) and the expected distance between the global model and the related optimum.

**Theorem 1** (Decomposition Theorem). *We consider $\boldsymbol{y}_{i,k}^t$ the local model of client $i$ after $k$ SGD steps initialized on model $\boldsymbol{\theta}^t$. We consider the vector $\boldsymbol{\xi}_i^t$ of the gradient noises $\boldsymbol{\xi}_{i,k}^t = [g_i(\boldsymbol{y}_{i,k}^t) - \nabla\mathcal{L}_i(\boldsymbol{y}_{i,k}^t)]$ satisfying Assumption 1, and $\boldsymbol{\Delta}_i^t = \sum_{k=0}^{K-1}\nabla\mathcal{L}_i(\boldsymbol{y}_{i,k}^t)$. We consider a client sampling scheme satisfying Definition 1, and such that $\text{Cov}\left[\omega_i(S_t), \omega_j(S_t)\right] = -\alpha p_i p_j$. For any given optimization round based on equation (3), the following equation holds:*

$$\mathbb{E}_t\left[\|\boldsymbol{\theta}^{t+1} - \boldsymbol{\theta}^*\|^2\right] = \|\boldsymbol{\theta}^t - \boldsymbol{\theta}^*\|^2 - 2\eta_g \underbrace{\langle\sum_{i=1}^{n}p_i\mathbb{E}_t\left[\boldsymbol{\theta}_i^{t+1} - \boldsymbol{\theta}^t\right], \boldsymbol{\theta}^* - \boldsymbol{\theta}^t\rangle}_{\textit{Direction drift}} + \eta_g^2\eta_l^2 Q(\boldsymbol{\theta}^t) \quad (7)$$

*with*

$$Q(\boldsymbol{\theta}^t) = \sum_{i=1}^{n} \nu_i \underbrace{\mathbb{E}_t \left[ \left\| \boldsymbol{\xi}_i^t \right\|^2 \right]}_{\substack{\textit{Gradient Estimator Noise} \\ =N_i(\boldsymbol{\theta}^t)}} + \sum_{i=1}^{n} \gamma_i \underbrace{\mathbb{E}_t \left[ \left\| \boldsymbol{\Delta}_i^t \right\|^2 \right]}_{\substack{\textit{Local Model Drift} \\ =L_i(\boldsymbol{\theta}^t)}} + (1-\alpha) \underbrace{\mathbb{E}_t \left[ \left\| \sum_{i=1}^{n} p_i \boldsymbol{\Delta}_i^t \right\|^2 \right]}_{\substack{\textit{Global Model Drift} \\ =G(\{p_i\}, \boldsymbol{\theta}^t)}}, \quad (8)$$

*where $\boldsymbol{\theta}^*$ is the optimum of the optimization problem (1), $\nu_i = \left[ \text{Var} \left[ \omega_i(S_t) \right] + p_i^2 \right]$, $\gamma_i = \text{Var} \left[ \omega_i(S_t) \right] + \alpha p_i^2$, and $\mathbb{E}_t \left[ X \right]$ is the expected value of $X$ conditioned on $\boldsymbol{\theta}^t$.*

The Decomposition Theorem (proof in Appendix B) shows that FL convergence is impacted by a client sampling through the quantities $\text{Var} \left[ \omega_i(S_t) \right]$ and $\alpha$, which both depend on the clients aggregation weights. These variables modulate three quantities, $N_i(\boldsymbol{\theta}^t)$, $L_i(\boldsymbol{\theta}^t)$ and $G(\{p_i\}, \boldsymbol{\theta}^t)$, representing respectively the stochastic gradient estimator noise, and the *local* and *global* model drift from the current initialization, which are independent from client sampling. In particular, the quality of clients data impacts the convergence speed through $N_i(\boldsymbol{\theta}^t)$. The Decomposition Theorem also provides a necessary condition for an optimization step to improve the current global model: the expected client contribution $\mathbb{E}_t \left[ \sum_{i=1}^{n} p_i \left[ \boldsymbol{\theta}_i^{t+1} - \boldsymbol{\theta}^t \right] \right]$ needs to be collinear with the global direction of the optimum $\boldsymbol{\theta}^* - \boldsymbol{\theta}^t$.

To further clarify the influence of client sampling on the FL convergence we introduce the following property (proof in Appendix A.1) introducing the quantity $\text{Var} \left[ \sum_{i=1}^{n} \omega_i(S_t) \right]$, the variance of the sum of the aggregation weights, and showing its relationship with the variance of the clients aggregation weights $\text{Var} \left[ \omega_i(S_t) \right]$, and with the covariance parameter $\alpha$.

**Property 1.** *For any client sampling, we have $0 \leq \alpha \leq 1$ and*

$$\text{Var} \left[ \sum_{i=1}^{n} \omega_i(S_t) \right] = \sum_{i=1}^{n} \text{Var} \left[ \omega_i(S_t) \right] - \alpha \left[ 1 - \sum_{i=1}^{n} p_i^2 \right]. \quad (9)$$

Since $0 \leq \alpha \leq 1$, the global model drift $G(\{p_i\}, \boldsymbol{\theta}^t)$ contributes positively to equation (8), and the term $Q(\boldsymbol{\theta}^t)$ is always positive. This means that $Q(\boldsymbol{\theta}^t)$ is not negligible, and an appropriate sampling scheme should be defined such that $Q(\boldsymbol{\theta}^t)$ is minimized. We note that the impact of a client sampling can always be mitigated by considering a smaller local learning rate $\eta_l$. Indeed, the client drift is proportional to $\eta_l$ while $Q(\boldsymbol{\theta}^t)$ is proportional to $\eta_l^2$.

## 3.2 ASYMPTOTIC FL CONVERGENCE WITH RESPECT TO CLIENT SAMPLING

To prove FL convergence with client sampling, our work relies on the following two assumptions (Wang et al., 2020a; Li et al., 2020a; Karimireddy et al., 2020; Haddadpour & Mahdavi, 2019; Wang et al., 2019a;b):

**Assumption 2** (Smoothness). *The clients local objective function is L-Lipschitz smooth, that is, $\forall i \in \{1, ..., n\}$, $\left\| \nabla \mathcal{L}_i(x) - \nabla \mathcal{L}_i(y) \right\| \leq L \left\| x - y \right\|$.*

**Assumption 3** (Bounded Dissimilarity ). *There exist constants $\beta^2 \geq 1$ and $\kappa^2 \geq 0$ such that for every combination of positive weights $\{w_i\}$ such that $\sum_{i=1}^{n} w_i = 1$, we have $\sum_{i=1}^{n} w_i \left\| \nabla \mathcal{L}_i(x) \right\|^2 \leq \beta^2 \left\| \nabla \mathcal{L}(x) \right\|^2 + \kappa^2$. If all the local loss functions are identical, then we have $\beta^2 = 1$ and $\kappa^2 = 0$.*

We formalize in the following theorem the relationship between the statistical properties of the client sampling scheme and the asymptotic convergence of FL (proof in Appendix C).

**Theorem 2** (FL convergence). *Let us consider a client sampling scheme satisfying Definition 1 and such that for $i \neq j$, $\text{Cov} \left[ \omega_i(S_t), \omega_j(S_t) \right] = -\alpha p_i p_j \leq 0$. Under Assumptions 1 to 3, and with sufficiently small local step size $\eta_l$, the following convergence bound holds:*

$$\frac{1}{T} \sum_{t=0}^{T-1} \mathbb{E} \left[ \left\| \nabla \mathcal{L}(\boldsymbol{\theta}^t) \right\|^2 \right] \leq \mathcal{O} \left( \frac{1}{\eta_g \eta_l KT} \right) + \mathcal{O} \left( \eta_l^2 (K-1) \sigma^2 \right) + \mathcal{O} \left( \eta_l^2 K(K-1) \kappa^2 \right)$$

$$+ \mathcal{O} \left( \eta_g \eta_l \left[ \Sigma + \sum_{i=1}^{n} p_i^2 \right] \sigma^2 \right) + \mathcal{O} \left( \eta_g \eta_l \gamma \left[ (K-1) \sigma^2 + K \kappa^2 \right] \right), \quad (10)$$

*where $K$ is the number of local SGD, and*

$$\Sigma = \sum_{i=1}^{n} \text{Var} \left[ \omega_i(S_t) \right] \ and \ \gamma = \sum_{i=1}^{n} \text{Var} \left[ \omega_i(S_t) \right] + \alpha \sum_{i=1}^{n} p_i^2. \tag{11}$$

We first observe that any client sampling scheme satisfying the assumptions of Theorem 2 converges to its optimum. Through $\Sigma$ and $\gamma$, equation (10) shows that our bound is proportional to the clients aggregation weights through the quantities $\text{Var} \left[ \omega_i(S_t) \right]$ and $\alpha$, which thus should be minimized. These terms are non-negative and are minimized and equal to zero only with full participation of the clients to every optimization round. Theorem 2 does not require the sum of the weights $\omega_i(S_t)$ to be equal to 1. Yet, for client sampling satisfying $\text{Var} \left[ \sum_{i=1}^{n} \omega_i(S_t) \right] = 0$, we get $\alpha \propto \Sigma$. Hence, choosing an optimal client sampling scheme amounts at choosing the client sampling with the smallest $\Sigma$. This aspect has been already suggested in Fraboni et al. (2021).

The convergence guarantee proposed in Theorem 2 extends the work of Wang et al. (2020a) where, in addition of considering FedAvg with clients performing $K$ vanilla SGD, we include a server learning rate $\eta_g$ and integrate client sampling (equation (3)). With full client participation ($\Sigma = \gamma = 0$) and $\eta_g = 1$, we retrieve the convergence guarantees of Wang et al. (2020a). Furthermore, our theoretical framework can be applied to any client sampling satisfying the conditions of Theorem 2. In turn, Theorem 2 holds for full client participation, MD sampling, Uniform sampling, as well as for the other client sampling schemes detailed in Appendix A. Finally, the proof of Theorem 2 is general enough to account for FL regularization methods (Li et al., 2020a; 2019; Acar et al., 2021), other SGD solvers (Kingma & Ba, 2015; Ward et al., 2019; Li & Orabona, 2019), and/or gradient compression/quantization (Reisizadeh et al., 2020; Basu et al., 2019; Wang et al., 2018). For all these applications, the conclusions drawn for client samplings satisfying the assumptions of Theorem 2 still hold.

### 3.3 APPLICATION TO CURRENT CLIENT SAMPLING SCHEMES

**MD sampling**. When using Table 1 to compute $\Sigma$ and $\gamma$ close-form we obtain:

$$\Sigma_{MD} = \frac{1}{m} \left[ 1 - \sum_{i=1}^{n} p_i^2 \right] \text{ and } \gamma_{MD} = \frac{1}{m}, \tag{12}$$

where we notice that $\Sigma_{MD} \leq \frac{1}{m} = \gamma_{MD}$. Therefore, one can obtain looser convergence guarantees than the ones of Theorem 2, independently from the amount of participating clients $n$ and set of clients importance $\{p_i\}$, while being inversely proportional to the amount of sampled clients $m$. The resulting bound shows that FL with MD sampling converges to its optimum for any FL application.

**Uniform sampling**. Contrarily to MD sampling, the stochastic aggregation weights of Uniform sampling do not sum to 1. As a result, we can provide FL scenarios diverging when coupled with Uniform sampling. Indeed, using Table 1 to compute $\Sigma$ and $\gamma$ close-form we obtain

$$\Sigma_U = \left[ \frac{n}{m} - 1 \right] \sum_{i=1}^{n} p_i^2 \text{ and } \gamma_U = \left[ 1 + \frac{1}{n-1} \right] \left[ \frac{n}{m} - 1 \right] \sum_{i=1}^{n} p_i^2, \tag{13}$$

where we notice that $\gamma_U = \left[ 1 + \frac{1}{n-1} \right] \Sigma_U$. Considering that $\sum_{i=1}^{n} p_i^2 \leq 1$, we have $\Sigma_U \leq \frac{n}{m} - 1$, which goes to infinity for large cohorts of clients and thus prevents FL with Uniform sampling to converge to its optimum. Indeed, the condition $\sum_{i=1}^{n} p_i^2 \leq 1$ accounts for every possible scenario of client importance $\{p_i\}$, including the very heterogeneous ones. In the special case where $p_i = 1/n$, we have $\sum_{i=1}^{n} p_i^2 = 1/n$, such that $\Sigma_U$ is inversely proportional to both $n$ and $m$. Such FL applications converge to the optimum of equation (1) for any configuration of $n$, $\{p_i\}$ and $m$.

Moreover, the comparison between the quantities $\Sigma$ and $\gamma$ for MD and Uniform sampling shows that Uniform sampling outperforms MD sampling when $p_i = 1/n$. More generally, Corollary 1 provides sufficient conditions with Theorem 2 for Uniform sampling to have better convergence guarantees than MD sampling (proof in Appendix C.7).

**Corollary 1.** *Uniform sampling has better convergence guarantees than MD sampling when $\Sigma_U \leq \Sigma_{MD}$, and $\gamma_U \leq \gamma_{MD}$ which is equivalent to*

$$\sum_{i=1}^{n} p_i^2 \leq \frac{1}{n-m+1}. \tag{14}$$

Corollary 1 can be related to $\text{Var}\left[\sum_{i=1}^{n} \omega_i(S_t)\right]$, the variance for the sum of the aggregation weights, which is always null for MD sampling, and different of 0 for Uniform sampling except when $p_i = 1/n$ for all the clients.

A last point of interest for the comparison between MD and Uniform sampling concerns the respective time complexity for selecting clients. Sampling with a Multinomial Distribution has time complexity $\mathcal{O}(n + m \log(n))$, where $\mathcal{O}(n)$ comes from building the probability density function to sample clients indices (Tang, 2019). This makes MD sampling difficult to compute or even intractable for large cohorts of clients. On the contrary sampling $m$ elements without replacement from $n$ states is a reservoir sampling problem and takes time complexity $\mathcal{O}(m(1 + \log(n/m))$(Li, 1994). In practice, clients either receive identical importance ($p_i = 1/n$) or an importance proportional to their data ratio, for which we may assume computation $p_i = \mathcal{O}(\frac{1}{n})$. As a result, for important amount $n$ of participating clients, Uniform sampling should be used as the default client sampling due to its lower time complexity. However, for small amount of clients and heterogeneous client importance, MD sampling should be used by default.

Due to space constraints, we only consider in this manuscript applying Theorem 2 to Uniform and MD sampling, which can also be applied to Binomial and Poisson Binomial sampling introduced in Section A, and satisfying our covariance assumption. To the best of our knowledge, we could only find *clustered sampling* introduced in Fraboni et al. (2021) not satisfying this assumption. Still, with minor changes, we provide for this sampling scheme a similar bound to the one of Theorem 2 (Appendix C.6), ultimately proving that clustered sampling improves MD sampling.

## 4 EXPERIMENTS

In this section we provide an experimental demonstration of the two main theorems of Section 3, by quantifying the correctness of Decomposition Theorem 1, and by verifying the convergence properties identified in Theorem 2.

### 4.1 SYNTHETIC EXPERIMENT

We consider a synthetic FL experiment where the clients models are defined by quadratic local loss functions. This choice enables closed-form computation of the global optimum, as well as of the expectation of the distance between global model and optimum during a single FL optimization step.

We consider $n = 10$ clients jointly learning a model with $m = 5$ of them sampled by the server at each optimization step. Each client local loss function is $\mathcal{L}_i(\boldsymbol{\theta}) = \frac{1}{2}\left\|\boldsymbol{\theta} - \boldsymbol{\theta}_i^*\right\|^2$ with optimum $\boldsymbol{\theta}_i^* \sim N(0_d, I_d)$, where $d = 20$ is the number of parameters in the model. The optimum of the global loss function of equation (1) is $\boldsymbol{\theta}^* = \sum_{i=1}^{n} p_i \boldsymbol{\theta}_i^*$. We consider varying $p_1$, the importance of client 1, while giving identical importance to the remaining clients, i.e.

$$p_1 = r \text{ and } p_i = (1 - r)/(n - 1), \ i \neq 1. \tag{15}$$

By varying $r$ between 0 and 1, we investigate scenarios with different level of heterogeneity for the clients importance. We perform full gradient descents, which makes client sampling the only source of randomness during optimization. Lastly, we only consider a single FL optimization step initialized with a global model $\boldsymbol{\theta}^0 \sim N(0_d, \boldsymbol{I}_d)$. We compare the average distance of 1000 simulations obtained after a single FL round to the theoretical one provided by the Decomposition Theorem 1 (see Appendix D for the closed form equation derived for the controlled example here proposed). The scenario above is tested in two different experimental settings: an iid one, where $\boldsymbol{\theta}_i^* = \boldsymbol{\theta}_1^*$, and a non-iid one, where every client local model is drawn independently.

The theoretical expected distance $\left\|\boldsymbol{\theta}^1 - \boldsymbol{\theta}^*\right\|^2$ is plotted for iid and non-iid case in Figure 1a and 1b. The corresponding experimental distance resulting from the 1000 simulations is shown in Figure 1c and 1d. We note that there is a close match between theoretical and experimental results, demonstrating the validity of Decomposition Theorem 1. Figure 1 highlights the following properties:

**Iid case** (Figure 1a and 1c). In the iid setting the expected difference obtained with MD sampling is equivalent to the one obtained when all the clients are considered (Full). This demonstrates that in the iid case the covariance parameter $\alpha$ does not affect FL convergence (Section 3). Moreover, Uniform sampling leads to generally poor convergence results, thus confirming that the variance of

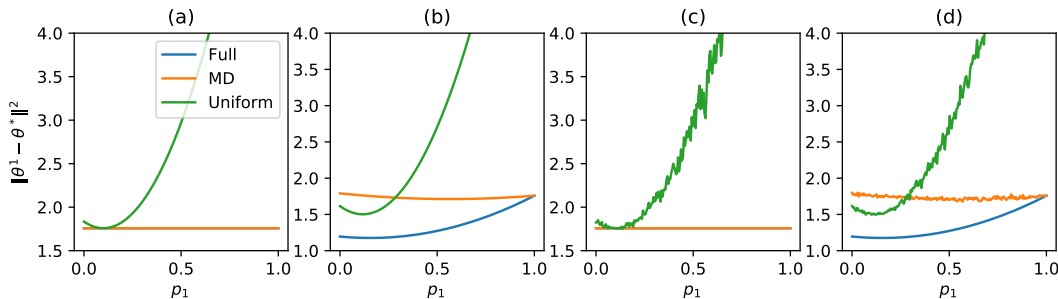

Figure 1: Illustration of the Decomposition Theorem 1 for the synthetic scenario described in Section 4.1 for $n = 10$ clients when sampling $m = 5$ of them. Panels (a) and (b) show the theoretical distances between global model and FL optimum obtained with the Decomposition Theorem 1 in respectively iid and non-iid settings. Panels (c) and (d) show the distances estimated experimentally when averaging over 1000 simulations for iid and non-iid settings. We consider $\eta_g = 1$, $\eta_l = 0.1$, and $K = 10$.

the sum of the clients stochastic weights, $\text{Var}_{S_t} \left[ \sum_{i=1}^n \omega_i(S_t) \right]$, negatively impacts FL convergence. More precisely, the only case where Uniform sampling is comparable to the other schemes is when $p_1 = 0.1$, which indeed corresponds to the special scenario in which $p_i = 1/n$, and thus the variance associated to Uniform sampling is zero (cfr. Table 1).

**Non iid case** (Figure 1b and 1d). The conclusions drawn for the iid case still hold when the local optima are not identical across clients. Moreover, we can now appreciate the impact of the covariance parameter $\alpha$ on the convergence speed. Independently from the choice for $p_1$, the minimum distance is obtained when considering all the clients. In this case we have both $\alpha = 0$ and $\text{Var}_{S_t} \left[ \sum_{i=1}^n \omega_i(S_t) \right] = 0$. When $p_i = 1/n = 0.1$, both MD and Uniform sampling have null variance $\text{Var}_{S_t} \left[ \sum_{i=1}^n \omega_i(S_t) \right]$. In this case, the negative impact of the covariance through the parameter $\alpha$ penalizes MD sampling. The penalization of the covariance term on MD sampling is still relevant when $p_1$ is close to the critical value of $0.1$. Finally, for large enough values of $p_1$, the variance term outweights the effect of the covariance, making MD better than Uniform sampling.

**Diverging Uniform sampling.** Consistently with Figure 1, in Appendix E we provide further examples showing that Uniform sampling can diverge in simple FL setting with heterogeneous client importance. We still consider $n = 10$ clients and sample $m = 5$ of them with $p_1 = r = 0.9$. When sampled, clients perform $K = 1$ SGD to show that the divergence does not come from the clients local work $K$. Figure 4 of Appendix E shows that for this scenario, FL with MD sampling converges while FL with Uniform sampling diverges both for iid and non-iid client data distribution.

## 4.2 EXPERIMENT ON REAL DATA

We study a LSTM model for next character prediction on the dataset of *The complete Works of William Shakespeare* (McMahan et al., 2017; Caldas et al., 2018). We use a two-layer LSTM classifier containing 100 hidden units with an 8 dimensional embedding layer. The model takes as an input a sequence of 80 characters, embeds each of the characters into a learned 8-dimensional space and outputs one character per training sample after 2 LSTM layers and a fully connected one.

When selected, a client performs $K = 50$ SGD steps on batches of size $B = 64$ with local learning rate $\eta_l = 1.5$. The server considers the clients local work with $\eta_g = 1$. We consider $n \in \{10, 20, 40, 80\}$ clients, and sample half of them at each FL optimization step. While for sake of interpretability we do not apply a decay to local and global learning rates, we note that our theory remains unchanged even in presence of a learning rate decay. In practice, for dataset with important heterogeneity, considering $\eta_g < 1$ can speed-up FL with a more stable convergence.

**Clients have identical importance** $[p_i = 1/n]$. We note that Uniform sampling consistently outperforms MD sampling due to the lower covariance parameter, while the improvement between the resulting convergence speed is inversely proportional to the number of participating clients $n$ (Figure 2a). This result confirms the derivations of Section 3.

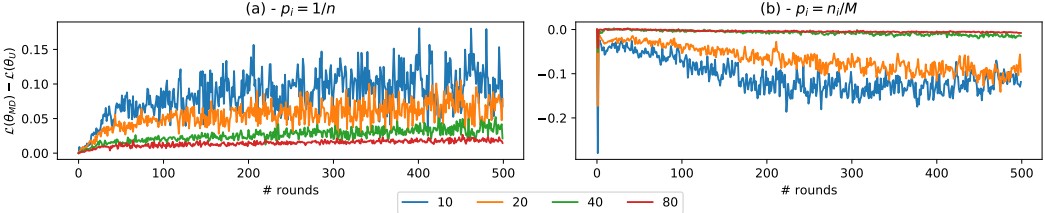

Figure 2: Difference between the convergence of the global losses resulting from MD and Uniform sampling when considering $n \in \{10, 20, 40, 80\}$ clients and sampling $m = n/2$ of them. In (a), clients have identical importance, i.e. $p_i = 1/n$. In (b), clients importance is proportional to their amount of data, i.e. $p_i = n_i/M$. Differences in global losses are averaged across 30 FL experiments with different model initialization (global losses are provided in Appendix E).

**Clients importance depends on the respective data ratio** $[p_i = n_i/M]$. In this experimental scenario the aggregation weights for Uniform sampling do not always sum to 1, thus leading to the slow-down of FL convergence. Hence, we see in Figure 2b that MD always outperforms Uniform sampling. This experiment shows that the impact on FL convergence of the variance of the sum of the stochastic aggregation weights is more relevant than the one due to the covariance parameter $\alpha$. We also note that the slow-down induced by the variance is reduced when more clients do participate. This is explained by the fact that the standard deviation of the clients data ratio is reduced with larger clients participation, e.g. $p_i = 1/10 \pm 0.13$ for $n = 10$ and $p_i = 1/80 \pm 0.017$ for $n = 80$. We thus conclude that the difference between the effects of MD and Uniform sampling is mitigated with a large number of participating clients (Figure 2b).

Additional experiments on Shakespeare are provided in Appendix E. We show the influence of the amount of sampled clients $m$ (Figure 6) and amount of local work $K$ (Figure 8) on the convergence speed of MD and Uniform sampling.

Finally, additional experiments on CIFAR10 (Krizhevsky, 2009) are provided in Appendix E, where we replicate the experimental scenario previously proposed in Fraboni et al. (2021). In these applications, 100 clients are partitioned using a Dirichlet distribution which provides federated scenarios with different level of heterogeneity. For all the experimental scenarios considered, both results and conclusions are in agreement with those here derived for the Shakespeare dataset.

## 5 CONCLUSION

In this work, we derive a novel Decomposition Theorem demonstrating the impact on FL convergence speed of any unbiased client sampling scheme following the assumptions of Section 2. Moreover, Theorem 2 highlights the asymptotic impact of client sampling on FL, and shows that the convergence speed is inversely proportional to both the sum of the variance of the stochastic aggregation weights, and to their covariance parameter $\alpha$. To the best of our knowledge, this work is the first one accounting for schemes where the sum of the weights is different from 1.

Thanks to our theory, we investigated MD and Uniform sampling from both theoretical and experimental standpoints. We established that when clients have approximately identical importance, i.e $p_i = 1/n$, Uniform outperforms MD sampling, due to the larger impact of the covariance term for the latter scheme. On the contrary, Uniform sampling is outperformed by MD sampling in more general cases, due to the slowdown induced by its stochastic aggregation weights not always summing to 1. Yet, in practical scenario with very large number of clients, MD sampling may be unpractical, and Uniform sampling could be preferred due to the more advantageous time complexity. Finally, while the contribution of this work is in the study of the impact of clients sampling on the global optimization objective, further extensions may focus on the analysis of the impact of clients selection method on individual users' performance, especially in presence of heterogeneity.

## ETHICS STATEMENT

We read and acknowledged the ICLR Code of Ethics.

The experimental correctness of our method was evaluated on publicly available datasets. Datasets were properly referenced and their construction can be found in the associated references.

This work investigates client sampling, an FL component already introduced in previous literature. Our work does not change the FL framework and thus does not propose additional client exposure.

There is no conflict of interest. The arguments used in this work are either theoretically proven, experimentally illustrated, or coming from peer-reviewed work.

Our method is centered around unbiased client sampling and thus does not induce bias in the global model.

## REPRODUCIBILITY STATEMENT

Every proposed theoretical result is carefully analyzed in Section 3 and proven in Appendix. Aggregation weight calculus can be found in Appendix A, Theorem 1 can be found in Appendix B, and Theorem 2 can be found in Appendix C.

The settings of the different experiments are specified in the paper and the code is provided. Furthermore, we use public datasets without any preprocessing.

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

# A   CLIENTS SAMPLING SCHEMES CALCULUS

In this section, we calculate for MD, Uniform, Poisson, and Binomial sampling the respective aggregation weight variance $\mathrm{Var}\left[\omega_i(S_t)\right]$, the covariance parameter $\alpha$ such that $\mathrm{Cov}\left[\omega_i(S_t)), \omega_j(S_t)\right] = -\alpha p_i p_j$, and the variance of the sum of weights $\mathrm{Var}\left[\sum_{i=1}^n \omega_i(S_t)\right]$. We also propose statistics for the parameter $N$, i.e. the amount of clients the server communicates with at an iteration:

$$N = \sum_{i=1}^n \mathbb{I}(i \in S_t). \tag{16}$$

## A.1   PROOF OF PROPERTY 1

*Proof.* **Covariance parameter**

$$\mathrm{Cov}\left[\omega_i(S_t), \omega_j(S_t)\right] = \mathbb{E}\left[\omega_i(S_t)\omega_j(S_t)\right] - p_i p_j \geq -p_i p_j \tag{17}$$

**Aggregation Weights Sum**

$$\mathrm{Var}\left[\sum_{i=1}^n \omega_i(S_t)\right] = \sum_{i=1}^n \mathrm{Var}\left[\omega_i(S_t)\right] + \sum_{i,j \neq i} \mathrm{Cov}\left[\omega_i(S_t), \omega_j(S_t)\right] \tag{18}$$

$$= \sum_{i=1}^n \mathrm{Var}\left[\omega_i(S_t)\right] - \alpha \sum_{i,j \neq i} p_i p_j \tag{19}$$

$$= \sum_{i=1}^n \left[\mathrm{Var}\left[\omega_i(S_t)\right] - \alpha p_i(1 - p_i)\right] \tag{20}$$

$$= \sum_{i=1}^n \mathrm{Var}\left[\omega_i(S_t)\right] - \alpha\left[1 - \sum_{i=1}^n p_i^2\right], \tag{21}$$

where we use $\sum_{i=1}^n p_i = 1$, equation (1), for the third and fourth equality. Using equation (20), we see that we can also get the following equality based on Decomposition Theorem 1:

$$\mathrm{Var}\left[\sum_{i=1}^n \omega_i(S_t)\right] = \sum_{i=1}^n \gamma_i - \alpha. \tag{22}$$

**Re-expressing $\alpha$.** Using equation (20), we get

$$\mathrm{Var}\left[\sum_{i=1}^n \omega_i(S_t)\right] = \sum_{i=1}^n \mathrm{Var}\left[\omega_i(S_t)\right] - \alpha\left[1 - \sum_{i=1}^n p_i^2\right], \tag{23}$$

which, with reordering, gives

$$\alpha = \frac{\sum_{i=1}^n \mathrm{Var}\left[\omega_i(S_t)\right] - \mathrm{Var}\left[\sum_{i=1}^n \omega_i(S_t)\right]}{1 - \sum_{i=1}^n p_i^2}. \tag{24}$$

$\square$

## A.2   NO SAMPLING SCHEME

When every client participate at an optimization round, we have $\omega_i(S_t) = p_i$ which gives $\mathrm{Var}_{S_t}\left[\omega_i(S_t)\right] = 0$, $\alpha = 0$, and $N = n$.

## A.3   MD SAMPLING

We recall equation (5),

$$\omega_i(S_t) = \frac{1}{m}\sum_{k=1}^m \mathbb{I}(l_k = i), \tag{25}$$

which gives

$$\mathbb{E}\left[\omega_i(S_t)\omega_j(S_t)\right] = \frac{1}{m^2}\sum_{k,l\neq k}\mathbb{E}\left[\mathbb{I}(l_k=i)\mathbb{I}(l_l=j)\right] + \frac{1}{m^2}\sum_{k=1}^{m}\mathbb{E}\left[\mathbb{I}(l_k=i)\mathbb{I}(l_k=j)\right] \tag{26}$$

$$= \frac{1}{m^2}\sum_{k,l\neq k}p_ip_j + \frac{1}{m^2}\sum_{k=1}^{m}\mathbb{E}\left[\mathbb{I}(l_k=i)\mathbb{I}(l_k=j)\right] \tag{27}$$

$$= \frac{m-1}{m}p_ip_j + \frac{1}{m}\mathbb{E}\left[\mathbb{I}(l=i)\mathbb{I}(l=j)\right] \tag{28}$$

**Variance**($i=j$). We get $\mathbb{E}\left[\mathbb{I}(l=i)\mathbb{I}(l=j)\right] = \mathbb{E}\left[\mathbb{I}(l=i)\right] = p_i$, which gives:

$$\mathrm{Var}\left[\omega_i(S_t)\right] = -\frac{1}{m}p_i^2 + \frac{1}{m}p_i \tag{29}$$

**Covariance**($i\neq j$). We get $\mathbb{E}\left[\mathbb{I}(l=i)\mathbb{I}(l=j)\right] = 0$, which gives:

$$\mathrm{Cov}\left[\omega_i(S_t),\omega_j(S_t)\right] = -\frac{1}{m}p_ip_j, \tag{30}$$

and by definition we get

$$\alpha = \frac{1}{m} \tag{31}$$

**Aggregation Weights Sum**. Using equation (29)) and (31) with Property 1, we get

$$\mathrm{Var}\left[\sum_{i=1}^{n}\omega_i(S_t)\right] = 0. \tag{32}$$

**Amount of clients**. Considering that $p(i\in S_t) = 1 - p(i\notin S_t) = 1 - (1-p_i)^m$, we get:

$$\mathbb{E}\left[N\right] = \sum_{i=1}^{n}\mathbb{P}(i\in S_t) = n - \sum_{i=1}^{n}(1-p_i)^m \leq m \tag{33}$$

## A.4 SAMPLING CLIENTS UNIFORMLY WITHOUT REPLACEMENT

We recall equation (6),

$$\omega_i(S_t) = \mathbb{I}(i\in S_t)\frac{n}{m}p_i. \tag{34}$$

**Variance**. We first calculate the probability for a client to be sampled, i.e.

$$\mathbb{P}(i\in S_t) = 1 - \mathbb{P}(i\notin S_t) = 1 - \frac{n-1}{n}...\frac{n-m}{n-m+1} = 1 - \frac{n-m}{n} = \frac{m}{n}. \tag{35}$$

Using equation (35), we have

$$\mathrm{Var}_{S_t}\left[\omega_i(S_t)\right] = \left[\frac{n}{m}p_i\right]^2\mathrm{Var}\left[\mathbb{I}(i\in S_t)\right] = \frac{n^2}{m^2}\frac{m}{n}(1-\frac{m}{n})p_i^2 = (\frac{n}{m}-1)p_i^2 \tag{36}$$

**Covariance**. We have

$$\mathbb{P}(\{i,j\}\in S_t) = \mathbb{P}(i\in S_t) + \mathbb{P}(j\in S_t) - \mathbb{P}(i\cup j\in S_t) \tag{37}$$

$$= \mathbb{P}(i\in S_t) + \mathbb{P}(j\in S_t) - (1-\mathbb{P}(\{i,j\}\notin S_t)), \tag{38}$$

and

$$\mathbb{P}(\{i,j\}\notin S_t) = \frac{n-2}{n}...\frac{n-m-1}{n-m+1} = \frac{(n-m)(n-m-1)}{n(n-1)}. \tag{39}$$

Substituting equation (35) and (39) in equation (38) gives

$$\mathbb{P}(\{i,j\} \in S_t) = 2\frac{m}{n} - 1 + \frac{(n-m)(n-m-1)}{n(n-1)} \tag{40}$$

$$= \frac{1}{n(n-1)} \left[ 2m(n-1) - n(n-1) + (n-m)(n-m-1) \right] \tag{41}$$

$$= \frac{m(m-1)}{n(n-1)}. \tag{42}$$

Hence, we can express the aggregation weights covariance as

$$\mathrm{Cov}\left[\omega_i(S_t), \omega_j(S_t)\right] = \frac{n^2}{m^2} \frac{m(m-1)}{n(n-1)} p_j p_k - p_j p_k, \tag{43}$$

which gives

$$\alpha = \frac{n-m}{m(n-1)}. \tag{44}$$

**Aggregation Weights Sum**. Combining equation (36) and (44) with Property 1 gives

$$\mathrm{Var}\left[\sum_{i=1}^{n} \omega_i(S_t)\right] = \sum_{i=1}^{n} \left[\frac{n}{m} - 1\right] p_i^2 - \frac{n-m}{m(n-1)} \sum_{i=1}^{n} p_i(1-p_i) \tag{45}$$

$$= \frac{n-m}{m(n-1)} \left[ n\sum_{i=1}^{n} p_i^2 - 1 \right], \tag{46}$$

where we retrieve $\mathrm{Var}\left[\sum_{i=1}^{n} \omega_i(S_t)\right] = 0$ for identical client importance, i.e. $\sum_{i=1}^{n} p_i^2 = \frac{1}{n}$.

**Amount of Clients**. $N = m$.

## A.5 POISSON BINOMIAL DISTRIBUTION

Clients are sampled according to a Bernoulli with a probability proportional to their importance $p_i$, i.e.

$$\omega_i(S_t) = \frac{1}{m}\mathbb{B}(mp_i). \tag{47}$$

Hence, only $m \geq p_{max}^{-1}$ can be sampled and we retrieve $\mathbb{E}\left[\omega_i(S_t)\right] = \frac{1}{m}mp_i = p_i$.

**Variance**.

$$\mathrm{Var}_{S_t}\left[\omega_i(S_t)\right] = \frac{1}{m^2}mp_i(1-mp_i) = \frac{1}{m}p_i(1-mp_i) \tag{48}$$

**Covariance**. Due to the independence of each stochastic weight, we also get:

$$\mathrm{Cov}\left[\omega_i(S_t), \omega_j(S_t)\right] = 0 \tag{49}$$

**Aggregation Weights Sum**. Using Property 1 we obtain

$$\mathrm{Var}\left[\sum_{i=1}^{n} \omega_i(S_t)\right] = \frac{1}{m} - \sum_{i=1}^{n} p_i^2. \tag{50}$$

**Amount of Clients**.

$$\mathbb{E}\left[N\right] = m \text{ and } \mathrm{Var}\left[N\right] = m - m^2 \sum_{i=1}^{n} p_i^2. \tag{51}$$

### A.6 BINOMIAL DISTRIBUTION

Clients are sampled according to a Bernoulli with identical sampling probability, i.e.

$$\omega_i(S_t) = \frac{n}{m}\mathbb{B}(\frac{m}{n})p_i. \tag{52}$$

Hence, we retrieve $\mathbb{E}\left[\omega_i(S_t)\right] = \frac{n}{m}\frac{m}{n}p_i = p_i$.

**Variance**.

$$\mathrm{Var}_{S_t}\left[\omega_i(S_t)\right] = \frac{n^2}{m^2}\frac{m}{n}(1-\frac{m}{n})p_i^2 = \frac{n-m}{m}p_i^2. \tag{53}$$

**Covariance**. Due to the independence of each stochastic weight, we have:

$$\mathrm{Cov}\left[\omega_i(S_t), \omega_j(S_t)\right] = 0. \tag{54}$$

**Aggregation Weights Sum**. Using Property 1 gives

$$\mathrm{Var}\left[\sum_{i=1}^{n}\omega_i(S_t)\right] = \frac{n-m}{m}\sum_{i=1}^{n}p_i^2. \tag{55}$$

**Amount of Clients**.

$$\mathbb{E}\left[N\right] = m \text{ and } \mathrm{Var}\left[N\right] = m - \frac{m^2}{n}. \tag{56}$$

### A.7 CLUSTERED SAMPLING

Clustered sampling (Fraboni et al., 2021) is a generalization of MD sampling where instead of sampling $m$ clients from the same distributions, $m$ clients are sampled from $m$ different distributions $\{W_k\}_{k=1}^{m}$ each of them privileging a different subset of clients. We denote by $r_{k,i}$ the probability of client $i$ to be sampled in distribution $k$. To satisfy Definition 1, the original work (Fraboni et al., 2021) provides the conditions:

$$\forall k \in \{1, ..., m\}, \sum_{i=1}^{n} r_{k,i} = 1 \text{ and } \forall i \in \{1, ..., n\}, \sum_{k=1}^{m} r_{k,i} = mp_i. \tag{57}$$

The clients aggregation weights remain identical to the one of MD sampling, i.e.

$$\omega_i(S_{Cl}) = \frac{1}{m}\sum_{k=1}^{K}\mathbb{I}(l_k = i), \tag{58}$$

where $\mathbb{I}(l_k = i)$ are still independently distributed but not identically.

We have

$$\mathbb{E}\left[\omega_i(S_t)\omega_j(S_t)\right] = \frac{1}{m^2}\sum_{k,l\neq k}\mathbb{E}\left[\mathbb{I}(l_k = i)\mathbb{I}(l_l = j)\right] + \frac{1}{m^2}\sum_{k=1}^{m}\mathbb{E}\left[\mathbb{I}(l_k = i)\mathbb{I}(l_k = j)\right] \tag{59}$$

$$= \frac{1}{m^2}\sum_{k,l\neq k}r_{k,i}r_{l,j} + \frac{1}{m^2}\sum_{k=1}^{m}\mathbb{E}\left[\mathbb{I}(l_k = i)\mathbb{I}(l_k = j)\right] \tag{60}$$

$$= p_i p_j - \frac{1}{m^2}\sum_{k=1}^{m}r_{k,i}r_{k,j} + \frac{1}{m^2}\sum_{k=1}^{m}\mathbb{E}\left[\mathbb{I}(l_k = i)\mathbb{I}(l_k = j)\right], \tag{61}$$

where we retrieve equation (28) when $r_{k,i} = p_i$.

**Variance** ($i = j$). We get $\mathbb{E}\left[\mathbb{I}(l_k = i)\mathbb{I}(l_k = j)\right] = \mathbb{E}\left[\mathbb{I}(l_k = i)\right] = r_{k,i}$, which gives:

$$\mathrm{Var}\left[\omega_i(S_{Cl})\right] = \frac{1}{m}p_i - \frac{1}{m^2}\sum_{k=1}^{m}r_{k,i}^2 \leq \mathrm{Var}\left[\omega_i(S_{MD})\right], \tag{62}$$

where the inequality comes from using the Cauchy-Schwartz inequality with equality if and only if all the $m$ distributions are identical, i.e. $r_{k,i} = p_i$.

**Covariance** $(i \neq j)$. We get $\mathbb{E}\left[\mathbb{I}(l_k = i)\mathbb{I}(l_k = j)\right] = 0$, which gives:

$$\text{Cov}\left[\omega_i(S_{Cl}), \omega_j(S_{Cl})\right] = -\frac{1}{m^2}\sum_{k=1}^{m} r_{k,i}r_{k,j} \leq \text{Cov}\left[\omega_i(S_{MD}), \omega_j(S_{MD})\right], \tag{63}$$

where the inequality comes from using the Cauchy-Schwartz inequality with equality if and only if all the $m$ distributions are identical, i.e. $r_{k,i} = p_i$.

**Aggregation Weights Sum**

$$\text{Var}\left[\sum_{i=1}^{n}\omega_i(S_{Cl})\right] = 0. \tag{64}$$

A.8 OPTIMAL SAMPLING

With optimal sampling (Chen et al., 2020b), clients are sampled according to a Bernoulli distribution with probability $q_i$, i.e.

$$\omega_i(S_t) = \frac{p_i}{q_i}\mathbb{B}(q_i). \tag{65}$$

Hence, we retrieve $\mathbb{E}\left[\omega_i(S_t)\right] = \frac{p_i}{q_i}q_i = p_i$.

**Variance**.

$$\text{Var}_{S_t}\left[\omega_i(S_t)\right] = \frac{1-q_i}{q_i}p_i^2. \tag{66}$$

**Covariance**. Due to the independence of each stochastic weight, we have:

$$\text{Cov}\left[\omega_i(S_t), \omega_j(S_t)\right] = 0. \tag{67}$$

**Aggregation Weights Sum**. Using Property 1 gives

$$\text{Var}\left[\sum_{i=1}^{n}\omega_i(S_t)\right] = \sum_{i=1}^{n}\frac{1-q_i}{q_i}p_i^2. \tag{68}$$

**Amount of Clients**.

$$\mathbb{E}\left[N\right] = \sum_{i=1}^{n}q_i \text{ and } \text{Var}\left[N\right] = \sum_{i=1}^{n}q_i(1-q_i). \tag{69}$$

Table 2: Common Notation Summary.

| Symbol | Description |
|---|---|
| $n$ | Number of clients. |
| $K$ | Number of local SGD. |
| $\eta_l$ | Local/Client learning rate. |
| $\eta_g$ | Global/Server learning rate. |
| $\tilde{\eta}$ | Effective learning rate, $\tilde{\eta} = \eta_l \eta_g$. |
| $\boldsymbol{\theta}^t$ | Global model at server iteration $t$. |
| $\boldsymbol{\theta}^*$ | Optimum of the federated loss function, equation (1). |
| $\boldsymbol{\theta}_i^{t+1}$ | Local update of client $i$ on model $\theta^t$. |
| $\boldsymbol{y}_{i,k}^t$ | Local model of client i after $k$ SGD ($y_{i,K}^t = \theta_i^{t+1}$ and $y_{i,0}^t = \theta^t$). |
| $p_i$ | Importance of client $i$ in the federated loss function, equation (1). |
| $m$ | Number of sampled clients . |
| $S_t$ | Set of participating clients considered at iteration $t$. |
| $\omega_i(S_t)$ | Aggregation weight for client $i$ given $S_t$. |
| $\alpha$ | Covariance parameter. |
| $\gamma_i$ | cf Section 3 |
| $\mathbb{E}_t[\cdot]$ | Expected value conditioned on $\theta^t$. |
| $\mathcal{L}(\cdot)$ | Federated loss function, equation 1 |
| $\mathcal{L}_i(\cdot)$ | Local loss function of client $i$. |
| $g_i(\cdot)$ | SGD. We have $\mathbb{E}_{\xi_i}[g_i(\cdot)] = \nabla\mathcal{L}_i(\cdot)$ with Assumption 1. |
| $\xi_i$ | Random batch of samples from client $i$ of size $B$. |
| $L$ | Lipschitz smoothness parameter, Assumption 2. |
| $\sigma^2$ | Bound on the variance of the stochastic gradients, Assumption 1. |
| $\beta, \kappa$ | Assumption 3 parameters on the clients gradient bounded dissimilarity. |

# B    DECOMPOSITION THEOREM 1

In Table 2, we provide the definition of the different notations used in this work. We also propose in Algorithm 1 the pseudo-code for FedAvg with aggregation scheme (3).

---

**Algorithm 1** Federated Learning based on equation (3)

---
1: The server sends to the $n$ clients the learning parameters $(K, \eta_l, B)$.
2: **for** $t \in \{0, ..., T-1\}$ **do**
3:     Sample a set of clients $S_t$ and get their aggregation weights $d_i(t)$.
4:     Send to clients in $S_t$ the current global model $\boldsymbol{\theta}^t$.
5:     Receive each sampled client contributions $c_i(t) = \boldsymbol{\theta}_i^{t+1} - \boldsymbol{\theta}^t$.
6:     Creates the new global model $\boldsymbol{\theta}^{t+1} = \boldsymbol{\theta}^t + \eta_g \sum_{i=1}^n d_i(t)c_i(t)$.
7: **end for**

---

## B.1    USEFUL LEMMA

We first introduce and prove the following useful lemma before introducing the proof of Theorem 1.

**Lemma 1.** *Let us consider $n$ vectors $\boldsymbol{x}_i, ..., \boldsymbol{x}_n$ and a client sampling satisfying $\mathbb{E}_{S_t}[\omega_i(S_t)] = p_i$ and $\mathrm{Cov}[\omega_i(S_t), \omega_j(S_t)] = -\alpha p_i p_j$. We have:*

$$\mathbb{E}_{S_t}\left[\left\|\sum_{i=1}^n \omega_i(S_t)\boldsymbol{x}_i\right\|^2\right] = \sum_{i=1}^n \gamma_i \|\boldsymbol{x}_i\|^2 + (1-\alpha)\left\|\sum_{i=1}^n p_i\boldsymbol{x}_i\right\|^2, \tag{70}$$

*where $\gamma_i = \mathrm{Var}_{S_t}[\omega_i(S_t)] + \alpha p_i^2$.*

*Proof.*

$$\mathbb{E}_{S_t}\left[\left\|\sum_{i=1}^{n}\omega_i(S_t)\boldsymbol{x}_i\right\|^2\right] = \sum_{i=1}^{n}\mathbb{E}_{S_t}\left[\omega_i(S_t)^2\right]\|\boldsymbol{x}_i\|^2 + \sum_{i=1}^{n}\sum_{\substack{j=1\\j\neq i}}^{n}\mathbb{E}_{S_t}\left[\omega_i(S_t)\omega_j(S_t)\right]\langle\boldsymbol{x}_i,\boldsymbol{x}_j\rangle. \tag{71}$$

In addition, we have:

$$\mathbb{E}_{S_t}\left[\omega_i(S_t)\omega_j(S_t)\right] = \mathrm{Cov}\left[\omega_i(S_t),\omega_j(S_t)\right] + p_ip_j = (-\alpha+1)p_ip_j, \tag{72}$$

where the last equality comes from the assumption on the client sampling covariance.

We also have:

$$\sum_{i=1}^{n}\sum_{\substack{j=1\\j\neq i}}^{n}\langle p_i\boldsymbol{x}_i, p_j\boldsymbol{x}_j\rangle = \left\|\sum_{i=1}^{n}p_i\boldsymbol{x}_i\right\|^2 - \sum_{i=1}^{n}p_i^2\|\boldsymbol{x}_i\|^2, \tag{73}$$

Substituting equation (72) and equation (73) in equation (71) gives:

$$\mathbb{E}_{S_t}\left[\left\|\sum_{i=1}^{n}\omega_i(S_t)\boldsymbol{x}_i\right\|^2\right] = \sum_{i=1}^{n}\left[\mathbb{E}_{S_t}\left[\omega_i(S_t)^2\right] - (-\alpha+1)p_i^2\right]\|\boldsymbol{x}_i\|^2 + (-\alpha+1)\left\|\sum_{i=1}^{n}p_i\boldsymbol{x}_i\right\|^2, \tag{74}$$

Considering that we have $\mathbb{E}_{S_t}\left[\omega_i(S_t)^2\right] = \mathrm{Var}\left[\omega_i(S_t)\right] + p_i^2$, we have :

$$\mathbb{E}_{S_t}\left[\omega_i(S_t)^2\right] + (\alpha-1)p_i^2 = \mathrm{Var}_{S_t}\left[\omega_i(S_t)\right] + \alpha p_i^2, \tag{75}$$

Substituting equation (75) in equation (74) completes the proof.

$\square$

## B.2 Proof of the Decomposition Theorem 1

*Proof.*

$$\mathbb{E}_t\left[\left\|\boldsymbol{\theta}^{t+1}-\boldsymbol{\theta}^*\right\|^2\right] = \mathbb{E}_t\left[\left\|(\boldsymbol{\theta}^{t+1}-\boldsymbol{\theta}^t)+(\boldsymbol{\theta}^t-\boldsymbol{\theta}^*)\right\|^2\right] \tag{76}$$

$$= \left\|\boldsymbol{\theta}^t-\boldsymbol{\theta}^*\right\|^2 + 2\langle\mathbb{E}_t\left[\boldsymbol{\theta}^{t+1}-\boldsymbol{\theta}^t\right],\boldsymbol{\theta}^t-\boldsymbol{\theta}^*\rangle + \mathbb{E}_t\left[\left\|\boldsymbol{\theta}^{t+1}-\boldsymbol{\theta}^t\right\|^2\right], \tag{77}$$

By construction (equation (3)), we have $\boldsymbol{\theta}^{t+1} - \boldsymbol{\theta}^t = \eta_g\sum_{i=1}^{n}\omega_i(S_t)(\boldsymbol{\theta}_i^{t+1}-\boldsymbol{\theta}^t)$. The sampling scheme follows Definition 1 which gives $\mathbb{E}_{S_t}\left[\boldsymbol{\theta}^{t+1}-\boldsymbol{\theta}^t\right] = \eta_g\sum_{i=1}^{n}p_i(\boldsymbol{\theta}_i^{t+1}-\boldsymbol{\theta}^t)$. Hence , we get:

$$\mathbb{E}_t\left[\left\|\boldsymbol{\theta}^{t+1}-\boldsymbol{\theta}^*\right\|^2\right] = \left\|\boldsymbol{\theta}^t-\boldsymbol{\theta}^*\right\|^2 + 2\eta_g\langle\sum_{i=1}^{n}p_i\,\mathbb{E}_t\left[\boldsymbol{\theta}_i^{t+1}-\boldsymbol{\theta}^t\right],\boldsymbol{\theta}^t-\boldsymbol{\theta}^*\rangle$$
$$+ \eta_g^2\,\mathbb{E}_t\underbrace{\left[\left\|\sum_{i=1}^{n}\omega_i(S_t)(\boldsymbol{\theta}_i^{t+1}-\boldsymbol{\theta}^t)\right\|^2\right]}_{=\eta_l^2 Q(\boldsymbol{\theta}^t)}. \tag{78}$$

Using Lemma 1, we express $Q(\boldsymbol{\theta}^t)$ as follow:

$$\eta_l^2 Q(\boldsymbol{\theta}^t) = \sum_{i=1}^{n}\gamma_i\,\mathbb{E}_t\left[\left\|\boldsymbol{\theta}_i^{t+1}-\boldsymbol{\theta}^t\right\|^2\right] + (1-\alpha)\,\mathbb{E}_t\left[\left\|\sum_{i=1}^{n}p_i(\boldsymbol{\theta}_i^{t+1}-\boldsymbol{\theta}^t)\right\|^2\right]. \tag{79}$$

By construction, we have $\boldsymbol{\theta}_i^{t+1} - \boldsymbol{\theta}^t = -\sum_{k=0}^{K-1}\eta_l g_i(y_{i,k}^t)$. Considering Assumption 1 gives

$$\mathbb{E}\left[\left\|\boldsymbol{\theta}_i^{t+1}-\boldsymbol{\theta}^t\right\|^2\right] = \eta_l^2\left[\mathbb{E}\left[\left\|\xi_i^t\right\|^2\right] + \mathbb{E}\left[\left\|\Delta_i^t\right\|^2\right]\right]. \tag{80}$$

Similarly, by using twice Assumption 1 for the global model drift, we have

$$\eta_l^2 \, \mathbb{E} \left[ \left\| \sum_{i=1}^n p_i d_i^t \right\|^2 \right] = \eta_l^2 \, \mathbb{E} \left[ \left\| \sum_{i=1}^n p_i \xi_i^t \right\|^2 \right] + \eta_l^2 \, \mathbb{E} \left[ \left\| \sum_{i=1}^n p_i \Delta_i^t \right\|^2 \right] \tag{81}$$

$$= \eta_l^2 \left[ \sum_{i=1}^n p_i^2 \, \mathbb{E} \left[ \left\| \xi_i^t \right\|^2 \right] + \mathbb{E} \left[ \left\| \sum_{i=1}^n p_i \Delta_i^t \right\|^2 \right] \right], \tag{82}$$

where, for the last equality, Assumption 1 gives $\mathbb{E} \left[ \langle \xi_i^t, \xi_j^t \rangle \right] = \mathbb{E} \left[ \xi_i^t \right]^T \mathbb{E} \left[ \xi_j^t \right] = 0$.

Combining equation (80) with equation (82) gives

$$Q(\boldsymbol{\theta}^t) = \sum_{i=1}^n \left[ \mathrm{Var} \left[ \omega_i(S_t) \right] + p_i^2 \right] \mathbb{E} \left[ \left\| \xi_i^t \right\|^2 \right]$$

$$+ \sum_{i=1}^n \gamma_i \, \mathbb{E} \left[ \left\| \Delta_i^t \right\|^2 \right] + (1 - \alpha) \, \mathbb{E} \left[ \left\| \sum_{i=1}^n p_i \Delta_i^t \right\|^2 \right]. \tag{83}$$

Substituting equation (83) in equation (78) completes the proof.

$\square$

### B.3 ADAPTATION TO CLUSTERED SAMPLING

Instead of Lemma 1 which requires $\mathrm{Cov} \left[ \omega_i(S_t), \omega_j(S_t) \right] = -\alpha p_i p_j$, we propose the following Lemma for clustered sampling expressed in function of MD sampling covariance parameter $\alpha_{MD}$ showing that a sufficient condition for MD sampling to perform as well as Clustered sampling is that all $\boldsymbol{x}_i$ are identical, or that all the distributions are identical, i.e. $r_{k,i} = p_i$.

**Lemma 2.** *Let us consider $n$ vectors $\boldsymbol{x}_i, ..., \boldsymbol{x}_n$ and a client sampling satisfying $\mathbb{E}_{S_t} \left[ \omega_i(S_t) \right] = p_i$ and $\mathrm{Cov} \left[ \omega_i(S_{Cl}), \omega_j(S_{Cl}) \right] = -\alpha p_i p_j$. We have:*

$$\mathbb{E}_{S_{Cl}} \left[ \left\| \sum_{i=1}^n \omega_i(S_{Cl}) \boldsymbol{x}_i \right\|^2 \right] \le \sum_{i=1}^n \gamma_i(MD) \left\| \boldsymbol{x}_i \right\|^2 + (1 - \alpha_{MD}) \left\| \sum_{i=1}^n p_i \boldsymbol{x}_i \right\|^2, \tag{84}$$

*where $\gamma_i(MD)$ and $\alpha_{MD}$ are the aggregation weights statistics of MD sampling. Equation (84) is an equality if and only if $\sum_{i=1}^n r_{k,i} \boldsymbol{x}_i = \sum_{j=1}^n r_{k,j} \boldsymbol{x}_j$.*

*Proof.* Substituting equation (62) in equation (71) gives

$$\mathbb{E}_{S_{Cl}} \left[ \left\| \sum_{i=1}^n \omega_i(S_{Cl}) \boldsymbol{x}_i \right\|^2 \right] = \sum_{i=1}^n \mathbb{E}_{S_{Cl}} \left[ \omega_i(S_{Cl})^2 \right] \left\| \boldsymbol{x}_i \right\|^2 + \sum_{i=1}^n \sum_{\substack{j=1 \\ j \ne i}}^n p_i p_j \langle \boldsymbol{x}_i, \boldsymbol{x}_j \rangle$$

$$- \frac{1}{m^2} \sum_{k=1}^m \sum_{i=1}^n \sum_{\substack{j=1 \\ j \ne i}}^n r_{k,i} r_{k,j} \langle \boldsymbol{x}_i, \boldsymbol{x}_j \rangle, \tag{85}$$

Substituting equation (73) in equation (71) gives:

$$\mathbb{E}_{S_{Cl}} \left[ \left\| \sum_{i=1}^n \omega_i(S_{Cl}) \boldsymbol{x}_i \right\|^2 \right] = \sum_{i=1}^n \mathbb{E}_{S_{Cl}} \left[ \omega_i(S_{Cl})^2 \right] \left\| \boldsymbol{x}_i \right\|^2 + \left\| \sum_{i=1}^n p_i \boldsymbol{x}_i \right\|^2 - \sum_{i=1}^n p_i^2 \left\| \boldsymbol{x}_i \right\|^2$$

$$- \frac{1}{m^2} \sum_{k=1}^m \left[ \left\| \sum_{i=1}^n r_{k,i} \boldsymbol{x}_i \right\|^2 - \sum_{i=1}^n r_{k,i}^2 \left\| \boldsymbol{x}_i \right\|^2 \right]. \tag{86}$$

With rearrangements and using equation (57) we get:

$$\mathbb{E}_{S_{Cl}} \left[ \left\| \sum_{i=1}^{n} \omega_i(S_{Cl}) \boldsymbol{x}_i \right\|^2 \right] = \sum_{i=1}^{n} \left[ \mathrm{Var}\left[\omega_i(S_{Cl})\right] + \frac{1}{m^2} \sum_{k=1}^{m} r_{k,i}^2 \right] \|\boldsymbol{x}_i\|^2 + \left\| \sum_{i=1}^{n} p_i \boldsymbol{x}_i \right\|^2$$

$$- \frac{1}{m^2} \sum_{k=1}^{m} \left\| \sum_{i=1}^{n} r_{k,i} \boldsymbol{x}_i \right\|^2 . \tag{87}$$

Using the expression of clustered sampling variance for the first term (equation (63)), and using Jensen's inequality on the third term completes the proof. Jensen's inequality is an equality if and only if $\sum_{i=1}^{n} r_{k,i} \boldsymbol{x}_i = \sum_{j=1}^{n} r_{k,j} \boldsymbol{x}_j$.

$\square$

## C    FL CONVERGENCE

Our work is based on the one of Wang et al. (2020a). We use the developed theoretical framework there proposed to prove Theorem 2. The focus of our work (and Theorem 2) is on FedAvg. Yet, the proof developed in this section, similarly to the one of Wang et al. (2020a), expresses $a_i$ in such a way they can account for a wide-range of regularization method on FedAvg, or optimizers different from Vanilla SGDone. This proof can easily be extended to account for different amount of local work from the clients (Wang et al., 2020a).

Before developing the proof of Theorem 2 in Section C.5, we introduce the notation we use in Section C.1, some useful lemmas in Section C.2 and Theorem 3 generalizing Theorem 2 in Section C.3.

### C.1    NOTATIONS

We define by $\boldsymbol{y}_{i,k}^t$ the local model of client $i$ after $k$ SGD steps initialized on $\boldsymbol{\theta}^t$, which enables us to also define the normalized stochastic gradients $\boldsymbol{d}_i^t$ and the normalized gradient $\boldsymbol{h}_i^t$ defined as

$$\boldsymbol{d}_i^t = \frac{1}{a_i} \sum_{k=0}^{K-1} a_{i,k} g_i(\boldsymbol{y}_{i,k}^t) \text{ and } \boldsymbol{h}_i^t = \frac{1}{a_i} \sum_{k=0}^{K-1} a_{i,k} \nabla \mathcal{L}_i(\boldsymbol{y}_{i,k}^t), \tag{88}$$

where $a_{i,k}$ is an arbitrary scalar applied by the client to its $k$th gradient, $\boldsymbol{a}_i = [a_{i,0}, .., a_{i,K-1}]^T$, and $a_i = \|\boldsymbol{a}_i\|_1$. In the special case of FedAvg, we have $\boldsymbol{a}_i = [1, ..., 1]$ and in the one of FedProx, we have $\boldsymbol{a}_i = [(1-\mu)^{K-1}, ..., 1]$ where $\mu$ is the FedProx regularization parameter.

With the formalism of equation (88), we can express a client contribution as $\boldsymbol{\theta}_i^{t+1} - \boldsymbol{\theta}^t = -\eta_l a_i \boldsymbol{d}_i^t$ and rewrite the server aggregation scheme defined in equation (3) as

$$\boldsymbol{\theta}^{t+1} - \boldsymbol{\theta}^t = -\eta_g \eta_l \sum_{i=1}^n \omega_i a_i \boldsymbol{d}_i^t, \tag{89}$$

which in expectation over the set of sampled clients $S_t$ gives

$$\mathbb{E}_{S_t} \left[ \boldsymbol{\theta}^{t+1} - \boldsymbol{\theta}^t \right] = -\tilde{\eta} \sum_{i=1}^n p_i a_i \boldsymbol{d}_i^t \tag{90}$$

$$= -\tilde{\eta} \underbrace{\left( \sum_{i=1}^n p_i a_i \right)}_{K_{eff}} \sum_{i=1}^n \underbrace{\left( \frac{p_i a_i}{\sum_{i=1}^n p_i a_i} \right)}_{w_i} \boldsymbol{d}_i^t. \tag{91}$$

We define the surrogate objective $\tilde{\mathcal{L}}(\boldsymbol{x}) = \sum_{i=1}^n w_i \mathcal{L}_i(\boldsymbol{x})$, where $\sum_{i=1}^n w_i = 1$.

In what follows, the norm used for $\boldsymbol{a}_i$ can either be L1, $\|\cdot\|_1$, or L2, $\|\cdot\|_2$. For other variables, the norm is always the euclidean one and $\|\cdot\|$ is used instead of $\|\cdot\|_2$. Also, regarding the client sampling metrics, for ease of writing, we use $\omega_i$ instead of $\omega_i(S_t)$ due to the independence of the client sampling statistics with respect to the current optimization round.

### C.2    USEFUL LEMMAS

**Lemma 3** (equation (87) in Wang et al. (2020a)). *Under Assumptions 1 to 3, we can prove*

$$\frac{1}{2} \sum_{i=1}^n w_i \mathbb{E}\left[ \left\| \nabla \mathcal{L}_i(\boldsymbol{\theta}^t) - \boldsymbol{h}_i^t \right\|^2 \right] \leq \frac{1}{2} \frac{\eta_l^2 L^2 \sigma^2}{1-R} \sum_{i=1}^n w_i \left( \|\boldsymbol{a}_i\|_2^2 - a_{i,-1}^2 \right)$$

$$+ \frac{R\beta^2}{2(1-R)} \mathbb{E}\left[ \left\| \nabla \tilde{\mathcal{L}}(\boldsymbol{\theta}^t) \right\|^2 \right] + \frac{R\kappa^2}{2(1-R)}, \tag{92}$$

*with $R = 2\eta_l^2 L^2 \max_i\{\|\boldsymbol{a}_i\|_1 (\|\boldsymbol{a}_i\|_1 - a_{i,-1})\}$ with a learning rate such that $R < 1$.*

*Proof.* The proof is in Section C.5 of Wang et al. (2020a).

The bound here provided is slightly tighter in term of numerical constants than the one of Wang et al. (2020a). Indeed, equation (70) in Wang et al. (2020a) uses the Jensen's inequality $\|\boldsymbol{a} + \boldsymbol{b}\|^2 \leq 2\|\boldsymbol{a}\|^2 + 2\|\boldsymbol{b}\|^2$ which could instead be obtained with:

$$\mathbb{E}\left[\left\|\sum_{s=0}^{k-1} a_{i,s} g_i(\boldsymbol{y}_{i,s}^t)\right\|^2\right] = \mathbb{E}\left[\left\|\sum_{s=0}^{k-1} a_{i,s}\left(g_i(\boldsymbol{y}_{i,s}^t) - \nabla\mathcal{L}_i(\boldsymbol{y}_{i,s}^t)\right)\right\|^2\right] + \mathbb{E}\left[\left\|\sum_{s=0}^{k-1} a_{i,s}\nabla\mathcal{L}_i(\boldsymbol{y}_{i,s}^t)\right\|^2\right],$$

$$\tag{93}$$

which uses Assumption 1, giving $\mathbb{E}\left[\langle\sum_{s=0}^{k-1} a_{i,s}\left(g_i(\boldsymbol{y}_{i,s}^t) - \nabla\mathcal{L}_i(\boldsymbol{y}_{i,s}^t)\right), \sum_{s=0}^{k-1} a_{i,s}\nabla\mathcal{L}_i(\boldsymbol{y}_{i,s}^t)\rangle\right] = 0$ with the same reasoning as for $U$ in equation (111). $\qquad\square$

**Lemma 4.** *Under Assumptions 1 to 3, we can prove*

$$\sum_{i=1}^n \gamma_i\,\mathbb{E}\left[\|a_i\boldsymbol{h}_i^t\|^2\right] \leq \frac{1}{1-R}\sigma^2 \sum_{i=1}^n \gamma_i\left(\|\boldsymbol{a}_i\|_2^2 - (a_{i,-1}^2)\right)$$

$$+ 2\left[\frac{R}{1-R} + 1\right]\left(\sum_{i=1}^n \gamma_i a_i^2\right)\left(\beta^2\,\mathbb{E}\left[\left\|\nabla\tilde{\mathcal{L}}(\boldsymbol{\theta}^t)\right\|^2\right] + \kappa^2\right), \tag{94}$$

*where $R' = 2\eta_l^2 L^2 \max_i\{\|a_i\|_1^2\} < 1$.*

*Proof.* Due to the definition of $\boldsymbol{h}_i^t$, we have:

$$\mathbb{E}\left[\|a_i\boldsymbol{h}_i^t\|^2\right] = a_i^2\,\mathbb{E}\left[\left\|\sum_{k=0}^{K-1} \frac{1}{a_i} a_{i,k}\nabla\mathcal{L}_i(\boldsymbol{y}_{i,k}^t)\right\|^2\right] \tag{95}$$

$$\leq a_i^2 \sum_{k=0}^{K-1} \frac{1}{a_i} a_{i,k}\,\mathbb{E}\left[\|\nabla\mathcal{L}_i(\boldsymbol{y}_{i,k}^t)\|^2\right]. \tag{96}$$

Using Jensen inequality, we have

$$\mathbb{E}\left[\|\nabla\mathcal{L}_i(\boldsymbol{y}_{i,k}^t)\|^2\right] \leq 2\,\mathbb{E}\left[\|\nabla\mathcal{L}_i(\boldsymbol{y}_{i,k}^t) - \nabla\mathcal{L}_i(\boldsymbol{\theta}^t)\|^2\right] + 2\,\mathbb{E}\left[\|\nabla\mathcal{L}_i(\boldsymbol{\theta}^t)\|^2\right] \tag{97}$$

$$\leq 2L^2\,\mathbb{E}\left[\|\boldsymbol{y}_{i,k}^t - \boldsymbol{\theta}^t\|^2\right] + 2\,\mathbb{E}\left[\|\nabla\mathcal{L}_i(\boldsymbol{\theta}^t)\|^2\right], \tag{98}$$

where the second equality comes from using Assumption 2.

Also, Section C.5 of Wang et al. (2020a) proves

$$\frac{1}{a_i}\sum_{k=0}^{K-1} a_{i,k}\,\mathbb{E}\left[\|\boldsymbol{y}_{i,k}^t - \boldsymbol{\theta}^t\|^2\right] \leq \frac{1}{1-R}\eta_l^2\sigma^2\left(\|\boldsymbol{a}_i\|_2^2 - (a_{i,-1}^2)\right) + \frac{1}{L^2}\frac{R}{1-R}\,\mathbb{E}\left[\|\nabla\mathcal{L}_i(\boldsymbol{\theta}^t)\|^2\right]. \tag{99}$$

Plugging equation (98) and then equation (99) in equation (96), we get:

$$\mathbb{E}\left[\|a_i\boldsymbol{h}_i^t\|^2\right] \leq a_i^2 \sum_{k=0}^{K-1} \frac{1}{a_i} a_{i,k}\left[2L^2\,\mathbb{E}\left[\|\boldsymbol{y}_{i,k}^t - \boldsymbol{\theta}^t\|^2\right] + 2\,\mathbb{E}\left[\|\nabla\mathcal{L}_i(\boldsymbol{\theta}^t)\|^2\right]\right] \tag{100}$$

$$= 2L^2 a_i^2 \sum_{k=0}^{K-1} \frac{1}{a_i} a_{i,k}\,\mathbb{E}\left[\|\boldsymbol{y}_{i,k}^t - \boldsymbol{\theta}^t\|^2\right] + 2a_i^2\,\mathbb{E}\left[\|\nabla\mathcal{L}_i(\boldsymbol{\theta}^t)\|^2\right] \tag{101}$$

$$\leq 2L^2 a_i^2\left[\frac{1}{1-R}\eta_l^2\sigma^2\left(\|\boldsymbol{a}_i\|_2^2 - (a_{i,-1}^2)\right) + \frac{1}{L^2}\frac{R}{1-R}\,\mathbb{E}\left[\|\nabla\mathcal{L}_i(\boldsymbol{\theta}^t)\|^2\right]\right]$$

$$+ 2a_i^2\,\mathbb{E}\left[\|\nabla\mathcal{L}_i(\boldsymbol{\theta}^t)\|^2\right] \tag{102}$$

$$\leq \frac{R'}{1-R}\sigma^2\left(\|\boldsymbol{a}_i\|_2^2 - (a_{i,-1}^2)\right) + 2a_i^2\left[\frac{R}{1-R} + 1\right]\mathbb{E}\left[\|\nabla\mathcal{L}_i(\boldsymbol{\theta}^t)\|^2\right]. \tag{103}$$

Summing over $n$ gives

$$\sum_{i=1}^{n} \gamma_i \, \mathbb{E}\left[\|a_i \boldsymbol{h}_i^t\|^2\right] \leq 2L^2 \frac{1}{1-R} \eta_l^2 \sigma^2 \sum_{i=1}^{n} \gamma_i a_i^2 \left(\|\boldsymbol{a}_i\|_2^2 - (a_{i,-1}^2)\right)$$
$$+ 2\left[\frac{R}{1-R} + 1\right] \sum_{i=1}^{n} \gamma_i a_i^2 \, \mathbb{E}\left[\|\nabla \mathcal{L}_i(\boldsymbol{\theta}^t)\|^2\right]. \tag{104}$$

Using Assumption 3 in equation (104) and $R' < 1$ completes the proof.

$\square$

### C.3 INTERMEDIARY THEOREM

**Theorem 3.** *The following inequality holds:*

$$\frac{1}{T} \sum_{t=0}^{T-1} \mathbb{E}\left[\left\|\nabla \tilde{\mathcal{L}}(\boldsymbol{\theta}^t)\right\|^2\right] \leq \mathcal{O}\left(\frac{1}{(1-\Omega)\tilde{\eta} \left(\sum_{i=1}^{n} p_i a_i\right) T}\right) + \mathcal{O}(\tilde{\eta} A' \sigma^2 \frac{1}{m}) + \mathcal{O}(\eta_l^2 \sigma^2 B')$$
$$+ \mathcal{O}(\eta_l^2 C' \kappa^2) + \mathcal{O}(\tilde{\eta} D' \sigma^2) + \mathcal{O}(\tilde{\eta} E' \kappa^2), \tag{105}$$

*where quantities A-E are defined in the following proof from equation (122) to equation (126).*

*Proof.* Clients local loss functions are $L$-Lipschitz smooth. Therefore, $\tilde{\mathcal{L}}$ is also $L$-Lipschitz smooth which gives

$$\mathbb{E}\left[\tilde{\mathcal{L}}(\boldsymbol{\theta}^{t+1}) - \tilde{\mathcal{L}}(\boldsymbol{\theta}^t)\right] \leq \underbrace{\mathbb{E}\left[\langle \nabla \tilde{\mathcal{L}}(\boldsymbol{\theta}^t), \boldsymbol{\theta}^{t+1} - \boldsymbol{\theta}^t \rangle\right]}_{T_1} + \frac{L}{2} \underbrace{\mathbb{E}\left[\|\boldsymbol{\theta}^{t+1} - \boldsymbol{\theta}^t\|^2\right]}_{T_2}, \tag{106}$$

where the expectation is taken over the subset of randomly sampled clients $S_t$ and the clients gradient estimator noises $\xi_i^t$. Please note that we use the notation $\mathbb{E}[\cdot]$ instead of $\mathbb{E}_{\{\xi_i^t\}, S_t}[\cdot]$ for ease of writing.

BOUNDING $T_1$

By conditioning on $\{\xi_i^t\}$ and using equation (91), we get:

$$T_1 = \mathbb{E}\left[\langle \nabla \tilde{\mathcal{L}}(\boldsymbol{\theta}^t), \mathbb{E}_{S_t}\left[\boldsymbol{\theta}^{t+1} - \boldsymbol{\theta}^t\right]\rangle\right] = -\tilde{\eta} K_{eff} \, \mathbb{E}\left[\langle \nabla \tilde{\mathcal{L}}(\boldsymbol{\theta}^t), \sum_{i=1}^{n} w_i \boldsymbol{h}_i^t \rangle\right], \tag{107}$$

which, using $2\langle a, b \rangle = \|a\|^2 + \|b\|^2 - \|a - b\|^2$ can be rewritten as:

$$T_1 = -\frac{1}{2}\tilde{\eta} K_{eff} \, \mathbb{E}\left[\left\|\nabla \tilde{\mathcal{L}}(\boldsymbol{\theta}^t)\right\|^2 + \left\|\sum_{i=1}^{n} w_i \boldsymbol{h}_i^t\right\|^2 - \left\|\nabla \tilde{\mathcal{L}}(\boldsymbol{\theta}^t) - \sum_{i=1}^{n} w_i \boldsymbol{h}_i^t\right\|^2\right]. \tag{108}$$

BOUNDING $T_2$

$$T_2|S_t = \tilde{\eta}^2 \, \mathbb{E}\left[\left\|\sum_{i=1}^n \omega_i a_i \boldsymbol{d}_i^t\right\|^2 |S_t\right] \tag{109}$$

$$= \tilde{\eta}^2 \, \mathbb{E}\left[\left\|\sum_{i=1}^n \omega_i a_i \left(\boldsymbol{d}_i^t - \boldsymbol{h}_i^t\right) + \sum_{i=1}^n \omega_i a_i \boldsymbol{h}_i^t\right\|^2 |S_t\right] \tag{110}$$

$$= \tilde{\eta}^2 \, \mathbb{E}\left[\left\|\sum_{i=1}^n \omega_i a_i \left(\boldsymbol{d}_i^t - \boldsymbol{h}_i^t\right)\right\|^2 |S_t\right] + \tilde{\eta}^2 \, \mathbb{E}\left[\left\|\sum_{i=1}^n \omega_i a_i \boldsymbol{h}_i^t\right\|^2 |S_t\right]$$

$$+ 2\tilde{\eta} \, \underbrace{\mathbb{E}\left[\langle\sum_{i=1}^n \omega_i a_i \left(\boldsymbol{d}_i^t - \boldsymbol{h}_i^t\right), \sum_{i=1}^n \omega_i a_i \boldsymbol{h}_i^t\rangle|S_t\right]}_{U}. \tag{111}$$

Using Assumption 1, we have $\mathbb{E}\left[\langle d_i^t - h_i^t, h_j^t\rangle\right] = 0$. Hence, we get $U = 0$ and can simplify $T_2$ as:

$$T_2 = \tilde{\eta}^2 \sum_{i=1}^n \mathbb{E}\left[\omega_i^2\right] a_i^2 \, \mathbb{E}\left[\left\|\boldsymbol{d}_i^t - \boldsymbol{h}_i^t\right\|^2\right] + \tilde{\eta}^2 \, \mathbb{E}\left[\left\|\sum_{i=1}^n \omega_i a_i \boldsymbol{h}_i^t\right\|^2\right]. \tag{112}$$

Using Lemma 1 on the second term, we get:

$$T_2 = \tilde{\eta}^2 \sum_{i=1}^n \mathbb{E}\left[\omega_i^2\right] a_i^2 \, \mathbb{E}\left[\left\|\boldsymbol{d}_i^t - \boldsymbol{h}_i^t\right\|^2\right] + \tilde{\eta}^2 \sum_{i=1}^n \gamma_i \, \mathbb{E}\left[\left\|a_i \boldsymbol{h}_i^t\right\|^2\right] + \tilde{\eta}^2 (1-\alpha) \, \mathbb{E}\left[\left\|\sum_{i=1}^n p_i a_i \boldsymbol{h}_i^t\right\|^2\right]. \tag{113}$$

Finally, by bounding the first term using Assumption 1, and noting that $p_i a_i = w_i K_{eff}$ for the second term, we get:

$$T_2 = \tilde{\eta}^2 \sum_{i=1}^n \mathbb{E}\left[\omega_i^2\right] \sum_{k=0}^{K-1} a_{i,k}^2 \, \mathbb{E}\left[\left\|g_i(\boldsymbol{y}_{i,k}^t) - \nabla\mathcal{L}_i(\boldsymbol{y}_{i,k}^t)\right\|^2\right]$$

$$+ \tilde{\eta}^2 \sum_{i=1}^n \gamma_i \, \mathbb{E}\left[\left\|a_i \boldsymbol{h}_i^t\right\|^2\right] + \tilde{\eta}^2 (1-\alpha) K_{eff}^2 \, \mathbb{E}\left[\left\|\sum_{i=1}^n w_i \boldsymbol{h}_i^t\right\|^2\right] \tag{114}$$

$$\leq \tilde{\eta}^2 \sum_{i=1}^n \mathbb{E}\left[\omega_i^2\right] \|\boldsymbol{a}_i\|_2^2 \sigma^2 + \tilde{\eta}^2 \sum_{i=1}^n \gamma_i \, \mathbb{E}\left[\left\|a_i \boldsymbol{h}_i^t\right\|^2\right] + \tilde{\eta}^2 (1-\alpha) K_{eff}^2 \, \mathbb{E}\left[\left\|\sum_{i=1}^n w_i \boldsymbol{h}_i^t\right\|^2\right]. \tag{115}$$

GOING BACK TO EQUATION (106)

Substituting equation (108) and equation (115) back in equation (106), we get:

$$\mathbb{E}\left[\tilde{\mathcal{L}}(\boldsymbol{\theta}^{t+1}) - \tilde{\mathcal{L}}(\boldsymbol{\theta}^t)\right] \leq -\frac{1}{2}\tilde{\eta} K_{eff} \left\|\nabla\tilde{\mathcal{L}}(\boldsymbol{\theta}^t)\right\|^2 + \frac{1}{2}\tilde{\eta} K_{eff} \, \mathbb{E}\left[\left\|\nabla\tilde{\mathcal{L}}(\boldsymbol{\theta}^t) - \sum_{i=1}^n w_i \boldsymbol{h}_i^t\right\|^2\right]$$

$$- \frac{1}{2}\tilde{\eta} K_{eff} \left[1 - L\tilde{\eta}(1-\alpha)K_{eff}\right] \mathbb{E}\left[\left\|\sum_{i=1}^n w_i \boldsymbol{h}_i^t\right\|^2\right]$$

$$+ \frac{L}{2}\tilde{\eta}^2 \sum_{i=1}^n \mathbb{E}\left[\omega_i^2\right] \|\boldsymbol{a}_i\|_2^2 \sigma^2 + \frac{L}{2}\tilde{\eta}^2 \sum_{i=1}^n \gamma_i \, \mathbb{E}\left[\left\|a_i \boldsymbol{h}_i^t\right\|^2\right], \tag{116}$$

We consider the learning rate to satisfy $1 - L\tilde{\eta}(1-\alpha)K_{eff} > 0$ such that we can simplify equation (116) as :

$$
\frac{\mathbb{E}\left[\tilde{\mathcal{L}}(\boldsymbol{\theta}^{t+1}) - \tilde{\mathcal{L}}(\boldsymbol{\theta}^t)\right]}{\tilde{\eta}K_{eff}} \leq -\frac{1}{2}\left\|\nabla\tilde{\mathcal{L}}(\boldsymbol{\theta}^t)\right\|^2 + \frac{1}{2}\mathbb{E}\left[\left\|\nabla\tilde{\mathcal{L}}(\boldsymbol{\theta}^t) - \sum_{i=1}^n w_i\boldsymbol{h}_i^t\right\|^2\right]
$$

$$
+ \frac{L}{2}\tilde{\eta}\frac{1}{K_{eff}}\sum_{i=1}^n \mathbb{E}\left[\omega_i^2\right]\|\boldsymbol{a}_i\|_2^2\sigma^2 + \frac{L}{2}\tilde{\eta}\frac{1}{K_{eff}}\sum_{i=1}^n \gamma_i\,\mathbb{E}\left[\|a_i\boldsymbol{h}_i^t\|^2\right] \quad (117)
$$

$$
\leq -\frac{1}{2}\left\|\nabla\tilde{\mathcal{L}}(\boldsymbol{\theta}^t)\right\|^2 + \frac{1}{2}\sum_{i=1}^n w_i\,\mathbb{E}\left[\|\nabla\mathcal{L}_i(\boldsymbol{\theta}^t) - \boldsymbol{h}_i^t\|^2\right]
$$

$$
+ \frac{L}{2}\tilde{\eta}\frac{1}{K_{eff}}\sum_{i=1}^n \mathbb{E}\left[\omega_i^2\right]\|\boldsymbol{a}_i\|_2^2\sigma^2 + \frac{L}{2}\tilde{\eta}\frac{1}{K_{eff}}\sum_{i=1}^n \gamma_i\,\mathbb{E}\left[\|a_i\boldsymbol{h}_i^t\|^2\right], \quad (118)
$$

where the last inequality uses the definition of the surrogate loss function $\tilde{\mathcal{L}}$ and the Jensen's inequality.

Using Lemma 4 and 3, we get:

$$
\frac{\mathbb{E}\left[\tilde{\mathcal{L}}(\boldsymbol{\theta}^{t+1}) - \tilde{\mathcal{L}}(\boldsymbol{\theta}^t)\right]}{\tilde{\eta}K_{eff}} \leq -\frac{1}{2}\left\|\nabla\tilde{\mathcal{L}}(\boldsymbol{\theta}^t)\right\|^2 + \frac{1}{2}\frac{\eta_l^2 L^2\sigma^2}{1-R}\sum_{i=1}^n w_i\left(\|\boldsymbol{a}_i\|_2^2 - a_{i,-1}^2\right)
$$

$$
+ \frac{R\beta^2}{2(1-R)}\mathbb{E}\left[\left\|\nabla\tilde{\mathcal{L}}(\boldsymbol{\theta}^t)\right\|^2\right] + \frac{R\kappa^2}{2(1-R)}
$$

$$
+ \frac{L}{2}\tilde{\eta}\frac{1}{K_{eff}}\left[\sum_{i=1}^n \mathbb{E}\left[\omega_i^2\right]\|\boldsymbol{a}_i\|_2^2 + \frac{1}{1-R}\sum_{i=1}^n \gamma_i\left(\|\boldsymbol{a}_i\|_2^2 - (a_{i,-1}^2)\right)\right]\sigma^2
$$

$$
+ L\tilde{\eta}\frac{1}{K_{eff}}\left[\frac{R}{1-R}+1\right]\left(\sum_{i=1}^n \gamma_i a_i^2\right)\left(\beta^2\,\mathbb{E}\left[\left\|\nabla\tilde{\mathcal{L}}(\boldsymbol{\theta}^t)\right\|^2\right] + \kappa^2\right). \tag{119}
$$

If we assume that $R \leq \frac{1}{2\beta^2+1}$, and considering that $\beta^2 \geq 1$, then we have $\frac{1}{1-R} \leq 1 + \frac{1}{2\beta^2} \leq \frac{3}{2}$, $\frac{R}{1-R} \leq \frac{1}{2}$, and $\frac{R\beta^2}{1-R} \leq \frac{1}{2\beta^2+1}(1 + \frac{1}{2\beta^2})\beta^2 = \frac{1}{2}$. We also define $\Omega = L\tilde{\eta}\frac{1}{K_{eff}}\frac{3}{2}\left(\sum_{i=1}^n \gamma_i a_i^2\right)\beta^2 \leq \frac{1}{2}$. Substituting these terms in equation (119) gives

$$
\frac{\mathbb{E}\left[\tilde{\mathcal{L}}(\boldsymbol{\theta}^{t+1}) - \tilde{\mathcal{L}}(\boldsymbol{\theta}^t)\right]}{\tilde{\eta}K_{eff}} \leq -\frac{1}{4}\left[1 - \Omega\right]\left\|\nabla\tilde{\mathcal{L}}(\boldsymbol{\theta}^t)\right\|^2 + \frac{3}{4}\eta_l^2 L^2\sigma^2\sum_{i=1}^n w_i\left(\|\boldsymbol{a}_i\|_2^2 - a_{i,-1}^2\right)
$$

$$
+ \frac{3}{2}\eta_l^2 L^2\max_i\{a_i(a_i - a_{i,-1})\}\kappa^2
$$

$$
+ \frac{L}{2}\tilde{\eta}\frac{1}{K_{eff}}\left[\sum_{i=1}^n \mathbb{E}\left[\omega_i^2\right]\|\boldsymbol{a}_i\|_2^2 + \frac{3}{2}\sum_{i=1}^n \gamma_i\left(\|\boldsymbol{a}_i\|_2^2 - (a_{i,-1}^2)\right)\right]\sigma^2
$$

$$
+ \frac{3}{2}L\tilde{\eta}\frac{1}{K_{eff}}\left(\sum_{i=1}^n \gamma_i a_i^2\right)\kappa^2. \tag{120}
$$

Averaging across all rounds, we get:

$$
\begin{aligned}
\frac{1-\Omega}{T}\sum_{t=0}^{T-1}\mathbb{E}\left[\left\|\nabla\tilde{\mathcal{L}}(\boldsymbol{\theta}^t)\right\|^2\right] \leq{}& 4\frac{\tilde{\mathcal{L}}(\theta^0)-\tilde{\mathcal{L}}(\theta^*)}{\tilde{\eta}K_{eff}T}+3\eta_l^2 L^2\sigma^2\sum_{i=1}^{n}w_i\left(\|\boldsymbol{a}_i\|_2^2-a_{i,-1}^2\right)\\
&+6\eta_l^2 L^2\max_i\{a_i(a_i-a_{i,-1})\}\kappa^2\\
&+L\tilde{\eta}\frac{1}{K_{eff}}\left[2\sum_{i=1}^{n}\mathbb{E}\left[\omega_i^2\right]\|\boldsymbol{a}_i\|_2^2+3\sum_{i=1}^{n}\gamma_i\left(\|\boldsymbol{a}_i\|_2^2-(a_{i,-1}^2)\right)\right]\sigma^2\\
&+6L\tilde{\eta}\frac{1}{K_{eff}}\left(\sum_{i=1}^{n}\gamma_i a_i^2\right)\kappa^2.
\end{aligned}
\tag{121}
$$

We define the following auxiliary variables

$$
A=m\frac{1}{K_{eff}}\sum_{i=1}^{n}\mathbb{E}\left[\omega_i^2\right]\|\boldsymbol{a}_i\|_2^2=m\frac{1}{\sum_{i=1}^{n}p_i a_i}\sum_{i=1}^{n}\left[\mathrm{Var}\left[\omega_i\right]+p_i^2\right]\|\boldsymbol{a}_i\|_2^2,
\tag{122}
$$

$$
B=\sum_{i=1}^{n}w_i\left(\|\boldsymbol{a}_i\|_2^2-a_{i,-1}^2\right)=\sum_{i=1}^{n}\frac{p_i a_i}{\sum_{j=1}^{n}p_j a_j}\left(\|\boldsymbol{a}_i\|_2^2-a_{i,-1}^2\right),
\tag{123}
$$

$$
C=\max_i\{a_i(a_i-a_{i,-1})\},
\tag{124}
$$

$$
D=\frac{1}{K_{eff}}\max_i\{a_i(a_i-a_{i,-1})\}\sum_{i=1}^{n}\gamma_i=\frac{1}{\sum_{i=1}^{n}p_i a_i}C\left(\sum_{i=1}^{n}\mathrm{Var}\left[\omega_i\right]+\alpha\sum_{i=1}^{n}p_i^2\right),
\tag{125}
$$

$$
E=\frac{1}{K_{eff}}\max_i\{a_i^2\}\left(\sum_{i=1}^{n}\gamma_i\right)=\frac{1}{\sum_{i=1}^{n}p_i a_i}\max_i\{a_i^2\}\left(\sum_{i=1}^{n}\mathrm{Var}\left[\omega_i\right]+\alpha\sum_{i=1}^{n}p_i^2\right).
\tag{126}
$$

We define for $A$ -$E$ the respective quantities $A'$-$E'$ such that $X'=\frac{1}{1-\Omega}X$. We have:

$$
\begin{aligned}
\frac{1}{T}\sum_{t=0}^{T-1}\mathbb{E}\left[\left\|\nabla\tilde{\mathcal{L}}(\boldsymbol{\theta}^t)\right\|^2\right]\leq{}& 4\frac{\tilde{\mathcal{L}}(\theta^0)-\tilde{\mathcal{L}}(\theta^*)}{(1-\Omega)\tilde{\eta}\left(\sum_{i=1}^{n}p_i a_i\right)T}+2L\tilde{\eta}A'\sigma^2\frac{1}{m}+3\eta_l^2 L^2\sigma^2 B'\\
&+6\eta_l^2 L^2 C'\kappa^2+3L\tilde{\eta}D\sigma^2+6L\tilde{\eta}E\kappa^2,
\end{aligned}
\tag{127}
$$

$\square$

## C.4 Synthesis of local learning rate $\eta_l$ conditions for Theorem 3

A sufficient bound on the local learning rate $\eta_l$ for constraints on $R$ for Lemma 3 and equation (119), and constraint on $R'$ for Lemma 4 to be satisfied is:

$$
2\left[2\beta^2+1\right]\eta_l^2 L^2\max_i\{\|a_i\|_1^2\}<1.
\tag{128}
$$

Constraints on equation (116) can be simplified as

$$
L\eta_g\eta_l(1-\alpha)K_{eff}<1.
\tag{129}
$$

Constraints on $\Omega$, equation (119), give

$$
3L\eta_g\eta_l\frac{1}{K_{eff}}\left(\sum_{i=1}^{n}\gamma_i a_i^2\right)\beta^2\leq 1.
\tag{130}
$$

## C.5 THEOREM 2

*Proof.* With FedAvg, every client performs vanilla SGD. As such, we have $a_{i,k} = 1$ which gives $a_i = K$ and $\|a_i\|_2 = \sqrt{K}$. In addition we consider a local learning rate $\eta_l$ such that $\Omega \leq \frac{1}{2}$ as such we can bound $A'$-$E'$ as $X' \leq 2X$.

Finally, considering that the variables $A$ to $E$ can be simplified as

$$A = m \sum_{i=1}^{n} \left[ \text{Var}\left[ \omega_i \right] + p_i^2 \right], \tag{131}$$

$$B = (K - 1), \tag{132}$$

$$C = K(K - 1), \tag{133}$$

$$D = (K - 1) \left( \sum_{i=1}^{n} \text{Var}\left[ \omega_i \right] + \alpha \sum_{i=1}^{n} p_i^2 \right), \tag{134}$$

and

$$E = K \left( \sum_{i=1}^{n} \text{Var}\left[ \omega_i \right] + \alpha \sum_{i=1}^{n} p_i^2 \right), \tag{135}$$

the convergence bound of Theorem 3 can be reduced to

$$\frac{1}{T} \sum_{t=0}^{T-1} \mathbb{E}\left[ \left\| \nabla \mathcal{L}(\boldsymbol{\theta}^t) \right\|^2 \right] \leq \mathcal{O}\left( \frac{1}{\eta_g \eta_l K T} \right) + \mathcal{O}\left( \eta_g \eta_l \sum_{i=1}^{n} \left[ \text{Var}\left[ \omega_i \right] + p_i^2 \right] \sigma^2 \right)$$
$$+ \mathcal{O}\left( \eta_l^2 (K - 1)\sigma^2 \right) + \mathcal{O}\left( \eta_l^2 K(K - 1)\kappa^2 \right)$$
$$+ \mathcal{O}\left( \eta_g \eta_l \left( \sum_{i=1}^{n} \text{Var}\left[ \omega_i \right] + \alpha \sum_{i=1}^{n} p_i^2 \right) \left[ (K - 1)\sigma^2 + K\kappa^2 \right] \right), \tag{136}$$

which completes the proof.

$\square$

$\Omega$ is proportional to $\sum_{i=1}^{n} \gamma_i = \sum_{i=1}^{n} \text{Var}\left[ \omega_i \right] + \alpha \sum_{i=1}^{n} p_i^2$. With full participation, we have $\Omega = 0$. However, with client sampling, all the terms in equation (136) are proportional with $\frac{1}{1-\Omega}$. Yet, we provide a looser bound in equation (136) independent from $\Omega$ as the conclusions drawn are identical. Through $\Omega$, $\sum_{i=1}^{n} \text{Var}\left[ \omega_i \right]$ and $\alpha$ needs to be minimized. This fact is already visible by inspection of the quantities $E$ and $F$.

We note that equation (136) depends on client sampling through $\sigma^2$, which is an indicator of the clients SGD quality, and $\kappa^2$, which depends on the clients data heterogeneity. In the special case where clients have the same data distribution and perform full gradient descent, based on the arguments discussed in the previous paragraph, we can still provide the following bound showing the influence of client sampling on the convergence speed, while highlighting the interest of minimizing the quantities $\sum_{i=1}^{n} \text{Var}\left[ \omega_i \right]$ and $\alpha$.

$$\frac{1}{T} \sum_{t=0}^{T-1} \mathbb{E}\left[ \left\| \nabla \mathcal{L}(\boldsymbol{\theta}^t) \right\|^2 \right] \leq \mathcal{O}\left( \frac{1}{(1 - \Omega)\eta_g \eta_l K T} \right), \tag{137}$$

When setting the server learning rate at 1, $\eta_g = 1$ with client full participation, i.e. $\text{Var}\left[ \omega_i \right] = \text{Var}\left[ \sum_{i=1}^{n} \omega_i \right] = \alpha = 0$ and $m = n$, we have $E = F = 0$ and can simplify $A$ to

$$A = n \sum_{i=1}^{n} p_i^2. \tag{138}$$

Therefore, the convergence guarantee we provide is $\frac{1}{\eta_l KT} + \eta_l \sum_{i=1}^n p_i^2 \sigma^2 + \eta_l^2 (K-1)\sigma^2 + \eta_l^2 K(K-1)\kappa^2$, which is identical to the one of Wang et al. (2020a) (equation (97) in their work), where $\sum_{i=1}^n p_i^2$ can be replaced by $1/n$ when clients have identical importance, i.e. $p_i = 1/n$.

In the special case, where we use $\eta_l = \sqrt{m/KT}$ (Wang et al., 2020a), we retrieve their asymptotic convergence bound $\frac{1}{\sqrt{mKT}} + \sqrt{\frac{m}{KT}} \sum_{i=1}^n p_i^2 \sigma^2 + \frac{m}{T}\sigma^2 + \frac{m}{T} K \kappa^2$.

## C.6 APPLICATION TO CLUSTERED SAMPLING

We adapt Theorem 2 to clustered sampling. Fraboni et al. (2021) prove the convergence of FL with clustered sampling by giving identical convergence guarantees to the one of FL with MD sampling. As a result, their convergence bound does not depend of the clients selection probability in the different clusters $r_{k,i}$. The authors' claim was that reducing the variance of the aggregation weights provides faster FL convergence, albeit only providing experimental proofs was provided to support this statement. Corollary 2 here proposed extends the theory of Fraboni et al. (2021) by theoretically demonstrating the influence of clustered sampling on the convergence rate. For easing the notation, Corollary 2 is adapted to FedAvg but can easily be extended to account for any local $a_i$ using the proof of Theorem 3 in Section C.3.

**Corollary 2.** *Even with no $\alpha$ such that $\mathrm{Cov}\left[\omega_i(S_t), \omega_j(S_t)\right] = -\alpha p_i p_j$, the bound of Theorem 2 still holds with B, C, and D defined as in Section C.3 and*

$$A = m\left[\frac{1}{m} - \frac{1}{m^2}\sum_{i=1}^n \sum_{k=1}^m r_{k,i}^2 + \sum_{i=1}^n p_i^2\right], \ E = \frac{1}{m}(K-1), \ and \ F = \frac{1}{m}K, \quad (139)$$

*where E and F are identical to the one for MD sampling and A is smaller than the one for Clustered sampling.*

*Proof.* The covariance property required for Theorem 3 is only used for Lemma 1. In the proof of Theorem 3, Lemma 1 is only used in equation (113). We can instead use Lemma 2 and keep the rest of the proof as it is in Section C.3. Therefore, the bound of Theorem 3 remains unchanged for clustered sampling where $E$ and $F$ use the aggregation weight statistics of MD sampling instead of clustered sampling. Statistics for MD sampling can be found in Section A.3 and give

$$\mathrm{Var}\left[\sum_{i=1}^n \omega_i(S_{MD})\right] = 0 \text{ and } \alpha_{MD} = \frac{1}{m}, \quad (140)$$

while the ones of clustered sampling in Section A.7 give

$$\sum_{i=1}^n \mathrm{Var}\left[\omega_i(S_{Cl})\right] = \frac{1}{m} - \frac{1}{m^2}\sum_{i=1}^n \sum_{k=1}^m r_{k,i}^2 \leq \sum_{i=1}^n \mathrm{Var}\left[\omega_i(S_{MD})\right]. \quad (141)$$

$\square$

## C.7 PROOF OF COROLLARY 1

*Proof.* Combining equation (29) with equation (36) gives

$$\Sigma_{MD} - \Sigma_U = \left[-\frac{1}{m}\sum_{i=1}^n p_i^2 + \frac{1}{m}\right] - \left(\frac{n}{m} - 1\right)\sum_{i=1}^n p_i^2 \quad (142)$$

$$= -\frac{1}{m}\left[(n - m + 1)\sum_{i=1}^n p_i^2 - 1\right]. \quad (143)$$

Therefore, we have

$$\Sigma_{MD} \leq \Sigma_U \Leftrightarrow \sum_{i=1}^n p_i^2 \leq \frac{1}{n - m + 1}. \quad (144)$$

Combining equation (31), (32), (44), and (46) gives

$$\gamma_{MD} - \gamma_U = \sum_{i=1}^{n} \text{Var}\left[\omega_i(S_{MD})\right] + \alpha_{MD} \sum_{i=1}^{n} p_i^2 - \left( \sum_{i=1}^{n} \text{Var}\left[\omega_i(S_U)\right] + \alpha_U \sum_{i=1}^{n} p_i^2 \right) \quad (145)$$

$$= \frac{1}{m} - \frac{n-m}{m(n-1)} n \sum_{i=1}^{n} p_i^2. \quad (146)$$

Therefore, we have

$$\gamma_{MD} \leq \gamma_U \Leftrightarrow \sum_{i=1}^{n} p_i^2 \leq \frac{1}{n-m} \frac{n-1}{n}. \quad (147)$$

Noting that

$$\frac{1}{n-m+1} - \frac{1}{n-m} \frac{n-1}{n} = \frac{-m+1}{n(n-m)(n-m+1)} \leq 0, \quad (148)$$

completes the proof.

$\square$

# D  QUADRATIC

*Proof.* For every client, we consider local loss functions such that:

$$\mathcal{L}_i(\boldsymbol{\theta}) = \frac{1}{2} \|\boldsymbol{\theta} - \boldsymbol{\theta}_i^*\|^2, \tag{149}$$

where $\boldsymbol{\theta}_i^*$ is client's $i$ local minimum. By taking the derivative of the global loss function, equation (1), with these clients local loss function, gives:

$$\boldsymbol{\theta}^* = \sum_{i=1}^{n} p_i \boldsymbol{\theta}_i^*. \tag{150}$$

$\boldsymbol{y}_{i,k}^t$ is the local model obtained for client $i$ after $k$ SGD and satisfies

$$\boldsymbol{y}_{i,k+1}^t = \boldsymbol{y}_{i,k}^t - \eta_l(\boldsymbol{y}_{i,k}^t - \boldsymbol{\theta}_i^*) = (1 - \eta_l)\boldsymbol{y}_{i,k}^t + \eta_l \boldsymbol{\theta}_i^*, \tag{151}$$

which, by induction, gives

$$\boldsymbol{\theta}_i^{t+1} = (1 - \eta_l)^K \boldsymbol{\theta}^t + \sum_{k=0}^{K-1} (1 - \eta_l)^k \eta_l \boldsymbol{\theta}_i^*. \tag{152}$$

We note that $\sum_{k=0}^{K-1}(1 - \eta_l)^k = [1 - (1 - \eta_l)^K]/\eta_l$. By defining $\phi = 1 - (1 - \eta_l)^K$, we get:

$$\boldsymbol{\theta}_i^{t+1} - \boldsymbol{\theta}^t = (1 - \eta_l)^K \boldsymbol{\theta}^t + \left[1 - (1 - \eta_l)^K\right] \boldsymbol{\theta}_i^* - \boldsymbol{\theta}^t = \phi(\boldsymbol{\theta}_i^* - \boldsymbol{\theta}^t). \tag{153}$$

Hence, we have

$$\sum_{i=1}^{n} p_i(\boldsymbol{\theta}_i^{t+1} - \boldsymbol{\theta}^t) = \phi(\boldsymbol{\theta}^* - \boldsymbol{\theta}^t). \tag{154}$$

Using Decomposition Theorem 1 leads to

$$\mathbb{E}\left[\|\boldsymbol{\theta}^{t+1} - \boldsymbol{\theta}^*\|^2\right] = (1 - 2\tilde{\eta})\,\mathbb{E}\left[\|\boldsymbol{\theta}^t - \boldsymbol{\theta}^*\|^2\right] + \eta_g^2 \sum_{i=1}^{n} \gamma_i \phi^2\,\mathbb{E}\left[\|\boldsymbol{\theta}^t - \boldsymbol{\theta}_i^*\|^2\right]$$

$$+ \eta_g^2(1 - \alpha)\phi^2\,\mathbb{E}\left[\|\boldsymbol{\theta}^t - \boldsymbol{\theta}^*\|^2\right]. \tag{155}$$

$\square$

# E    ADDITIONAL EXPERIMENTS

## E.1    SYNTHETIC EXPERIMENT

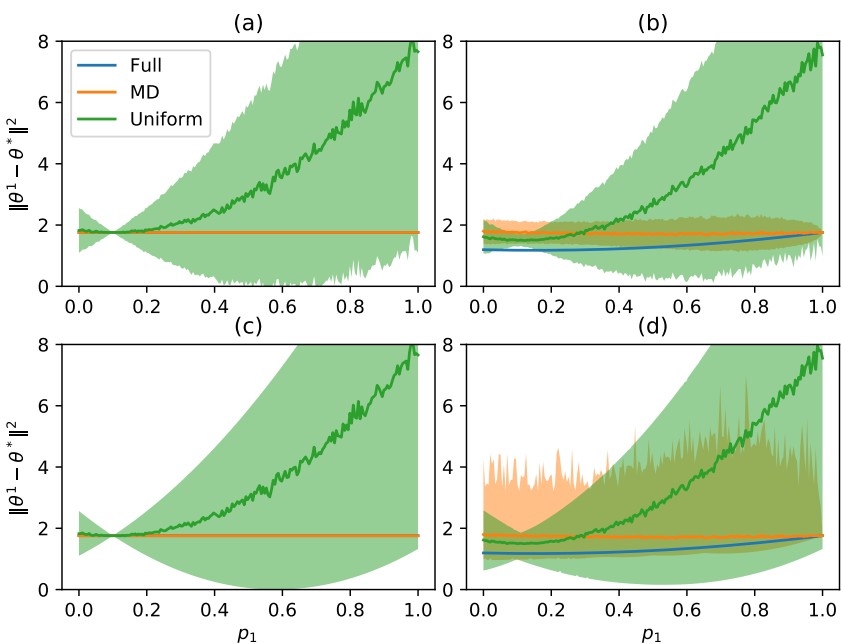

Figure 3: Additional plots for Figure 1. Illustration of the Decomposition Theorem 1 for the synthetic scenario described in Section 4.1 for $n = 10$ clients when sampling $m = 5$ of them. Panels (a) to (d) show the distances estimated experimentally when averaging over 1000 simulations for iid (Panels (a) and (c)), and non-iid settings, (Panels (b) and (d)), and with the associated standard deviation, (Panels (a) and (b)), and minimum and maximum distances, (Panels (c) and (d)). We consider $\eta_g = 1$, $\eta_l = 0.1$, and $K = 10$.

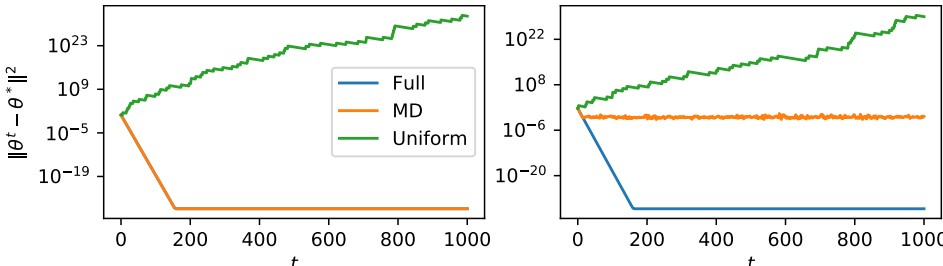

Figure 4: We consider the synthetic scenario described in Section 4.1 for $n = 100$ clients when sampling $m = 5$ and $r = 0.9$. We show the distances estimated experimentally when averaging over 1000 simulations for iid (Panel (a)), and non-iid settings (Panel (b)). We consider $\eta_g = 1$, $\eta_l = 0.2$, and $K = 1$. With $K = 1$, we show that the divergence does not come from asking too much work to the local clients. Uniform sampling divergence can be prevented by lowering the local learning rate $\eta_l$ (consistently with Theorem 2).

## E.2    SHAKESPEARE DATASET

The client local learning rate $\eta_l$ is selected in $\{0.1, 0.5, 1., 1.5, 2., 2.5\}$ minimizing FedAvg with full participation, and $n = 80$ training loss at the end of the learning process.

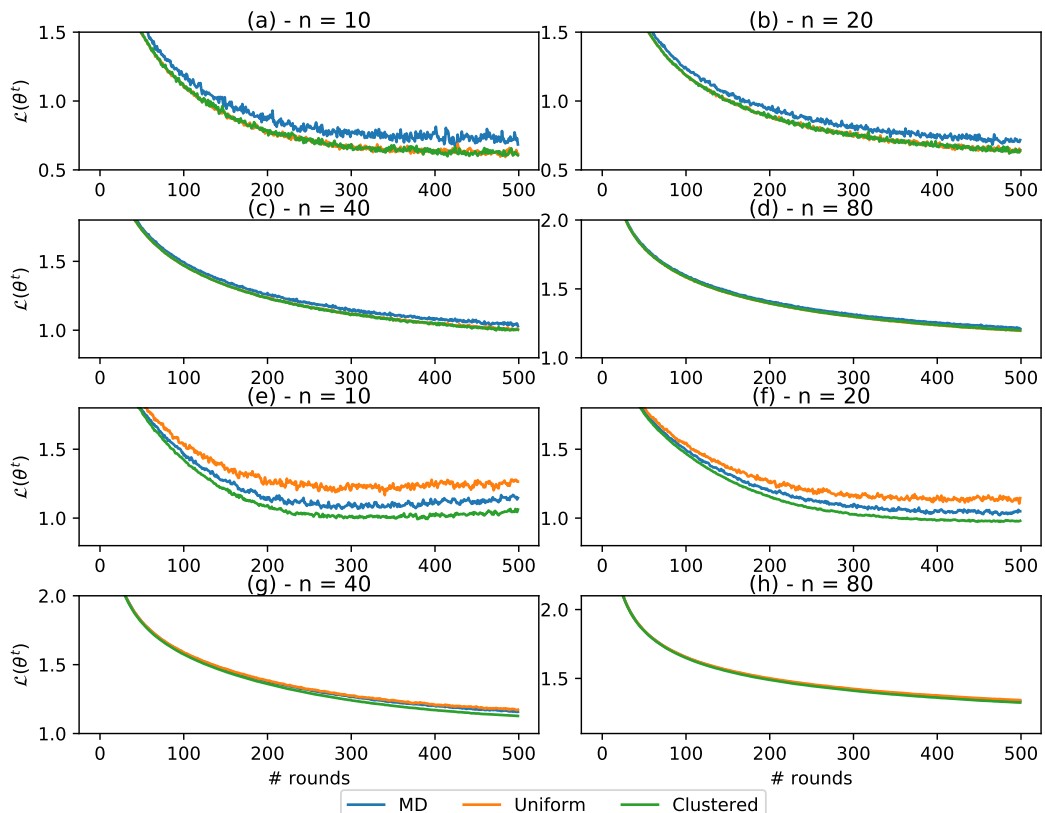

Figure 5: Convergence speed of the global loss with MD sampling and Uniform sampling when considering $n = 10$ ((a) and (e)), $n = 20$ ((b) and (f)), $n = 40$ ((c) and (g)), and $n = 80$ ((d) and (h)), while sampling $m = n/2$ of them. In (a-d) , clients have identical importance, i.e. $p_i = 1/n$, and, in (e-h), their importance is proportional to their amount of data, i.e. $p_i = n_i/M$. Global losses are estimated on 30 different model initialization.

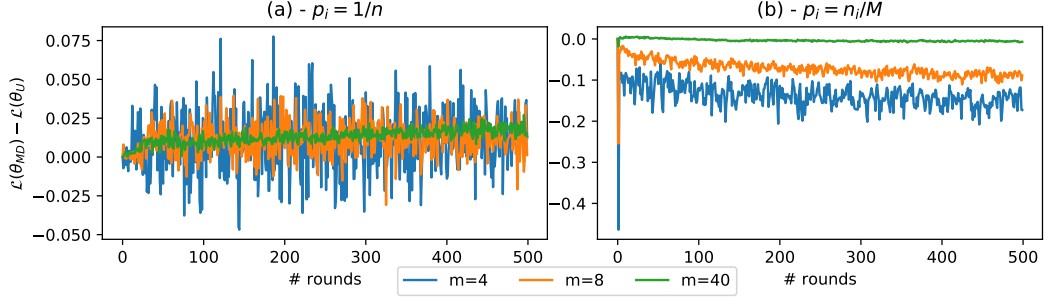

Figure 6: Difference between the convergence of the global losses resulting from MD and Uniform sampling when considering $n = 80$ clients and sampling $m \in \{4, 8, 40\}$ of them while clients perform $K = 50$ SGD steps . In (a), clients have identical importance, i.e. $p_i = 1/n$. In (b), clients importance is proportional to their amount of data, i.e. $p_i = n_i/M$. Differences in global losses are averaged across 15 FL experiments with different model initialization (global losses are provided in Figure 7).

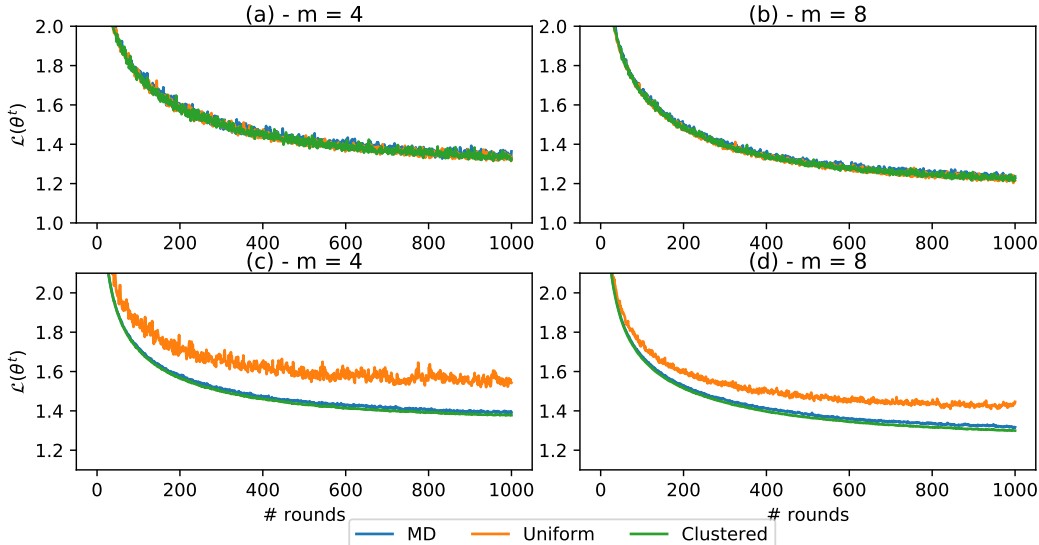

Figure 7: Convergence speed of the global loss with MD sampling and Uniform sampling when considering $n = 80$ clients while sampling $m = 4$ ((a) and (c)), and $m = 8$ ((b) and (d)) while clients perform $K = 50$ SGD steps. In (a-b) , clients have identical importance, i.e. $p_i = 1/n$, and, in (d-f), their importance is proportional to their amount of data, i.e. $p_i = n_i/M$. Global losses are estimated on 15 different model initialization.

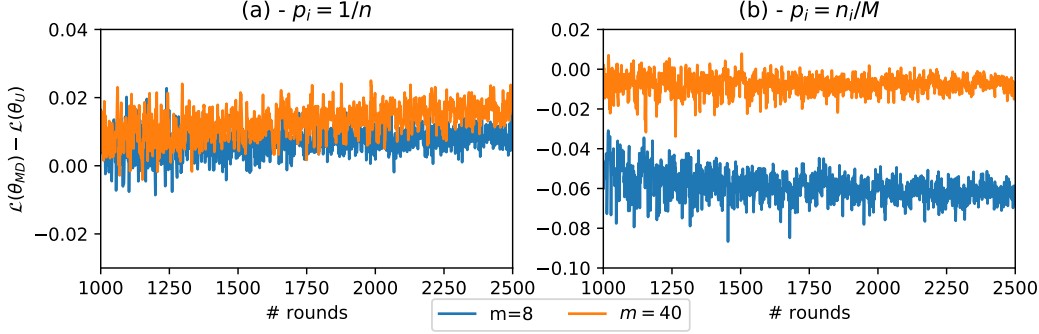

Figure 8: Difference between the convergence of the global losses resulting from MD and Uniform sampling when considering $n = 80$ clients and sampling $m \in \{8, 40\}$ of them while clients perform $K = 1$ SGD step. In (a), clients have identical importance, i.e. $p_i = 1/n$. In (b), clients importance is proportional to their amount of data, i.e. $p_i = n_i/M$. Differences in global losses are averaged across 15 FL experiments with different model initialization (global losses are provided in Figure 9).

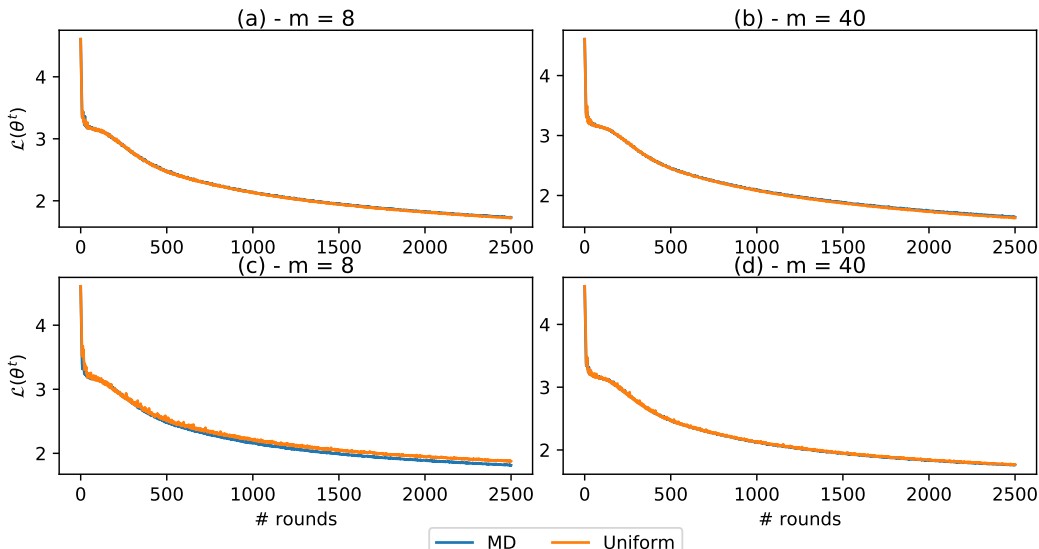

Figure 9: Convergence speed of the global loss with MD sampling and Uniform sampling when considering $n = 80$ clients while sampling $m = 4$ ((a) and (d)), $m = 8$ ((b) and (e)), $m = 40$ ((c) and (f)) while clients perform $K = 1$ SGD steps. In (a-c) , clients have identical importance, i.e. $p_i = 1/n$, and, in (d-f), their importance is proportional to their amount of data, i.e. $p_i = n_i/M$. Global losses are estimated on 15 different model initialization.

### E.3 CIFAR10 DATASET

We consider the experimental scenario used to prove the experimental correctness of clustered sampling in (Fraboni et al., 2021) on CIFAR10 (Krizhevsky, 2009). The dataset is partitioned in $n = 100$ clients using a Dirichlet distribution with parameter $\alpha = 0.1$ as proposed in Harry Hsu et al. (2019). 10, 30, 30, 20 and 10 clients have respectively 100, 250, 500, 750, and 1000 training samples, and testing samples amounting to a fifth of their training size.

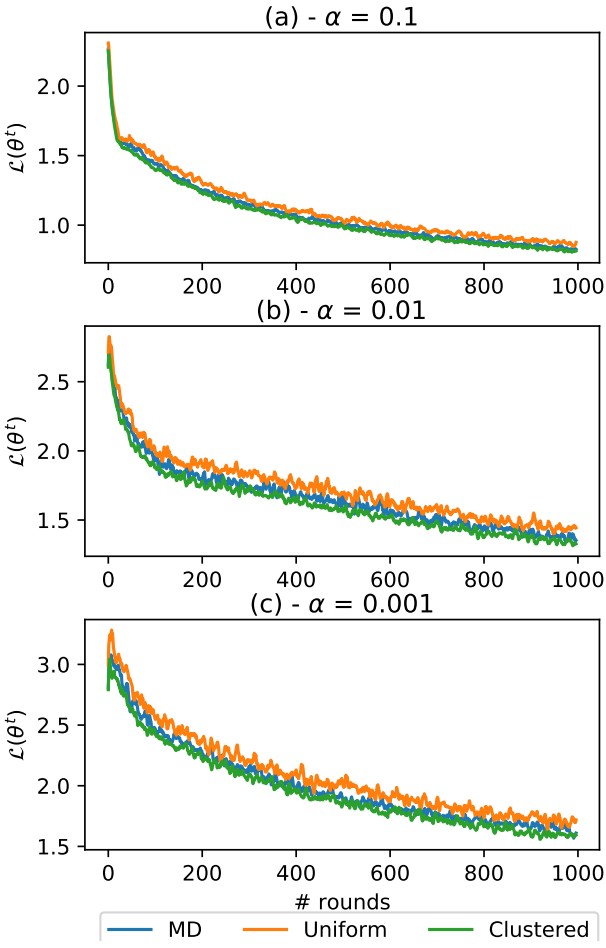

Figure 10: Convergence speed of the global loss with MD sampling and Uniform sampling when considering $n = 100$ clients, while sampling $m = 10$ of them. Clients are partitioned using a Dirichlet distribution with parameter $\alpha = 0.1$ (a), $\alpha = 0.01$ (b), and $\alpha = 0.001$ (c). Global losses are estimated on 30 different model initialization.

