# OpenReview forum: "On the Impact of Client Sampling on Federated Learning Convergence"
_ICLR.cc/2022/Conference — ICLR 2022 Submitted_

### Official Review · Reviewer_Sg3m · 2021-11-01

**Correctness:** 4
**Technical Novelty And Significance:** 3
**Empirical Novelty And Significance:** 2
**Recommendation:** 5
**Confidence:** 4

**Main Review:**

Structure.

This paper is well-written and it has a clear structure and narration. All assumptions are described and all variables are well defined. Proofs are also well-written and it is easy for a reader to follow. Proofs seem to be sound.

However, in this paper pseudo-code of the considered algorithm is not provided. This might lead to confusion. It can be helpful for readers to have pseudo-code for better understanding.

Authors use the phrase "after K SGD". Does it mean that K SGD steps? This phrase is confusing.

Assumptions.

The first assumption in this paper is Unbiased Gradient and Bounded Variance.

Assumption 1 (Unbiased Gradient and Bounded Variance). Every client stochastic gradient $g_{i}(\boldsymbol{x} \mid B)$ of a model $\boldsymbol{x}$ evaluated on batch $B$ is an unbiased estimator of the local gradient.

 We thus have $\mathbb{E}_{B}\left[{\xi}_{i}(B)\right]=0$ and $0 \leq \mathbb{E}_{B}\left[\left\|{\xi}_{i}(B)\right\|^{2}\right] \leq \sigma^{2}$, with ${\xi}_{i}(B)=g_{i}({x} \mid B)-\nabla \mathcal{L}_{i}({x})$

In https://arxiv.org/pdf/2002.03329.pdf it was shown that this assumption is limited and the simple 1-D counterexample is provided in proposition 1. Instead of using the Bounded Variance assumption, it is better to use the Expected Smoothness assumption:

Assumption (Expected smoothness). The second moment of the stochastic gradient satisfies
$$
\mathbb{E}\left[\|g(x)\|^{2}\right] \leq 2 A\left(f(x)-f^{\mathrm{inf}}\right)+B \cdot\|\nabla f(x)\|^{2}+C
$$
for some $A, B, C \geq 0$ and all $x \in \mathbb{R}^{d}$ and function $f$ is bounded from below by an infimum $ f^{\text {inf }} \in \mathbb{R}$.

Another questionable assumption is Assumption 3 (Bounded Dissimilarity ).
There exists constants $\beta^{2} \geq 1$ and $\kappa^{2} \geq 0$ such that for every combination of positive weights $\left\{w_{i}\right\}$ such that $\sum_{i=1}^{n} w_{i}=1$, we have $\sum_{i=1}^{n} w_{i}\left\|\nabla \mathcal{L}_{i}(x)\right\|^{2} \leq$ $\beta^{2}\|\nabla \mathcal{L}(x)\|^{2}+\kappa^{2} .$ If all the local loss functions are identical, then we have $\beta^{2}=1$ and $\kappa^{2}=0$.

This means that analysis does not cover arbitrary heterogeneous cases. In https://arxiv.org/pdf/1910.14425.pdf it was shown that this assumption is also limited. It is described in Remark 3.

Theory.

In this paper, there are two theorems. The first one is the decomposition theorem. This theorem provides several insights and it is an interesting result, which can lead to further improvement. However, in the statement, we have the phrase "after K SGD", which is not clear. However, FedAvg can be modeled not only by the local SGD method. There are many other approaches such as Federated Random Reshuffling (https://arxiv.org/abs/2102.06704, https://arxiv.org/pdf/2110.10342.pdf). It might be interesting to formulate similar theorems for other methods.

The second theorem provides convergence guarantees for local SGD. However, there is no epsilon complexity and comparison with other results. It might be useful for readers to have a table with different bounds. In http://proceedings.mlr.press/v130/gorbunov21a/gorbunov21a.pdf a lot of options and analyses are compared in Table 2.

Additionally, bounds for strongly convex and general convex cases are also desirable.

Experiments.

In this section the comparison between Multinomial and Uniform Distribution. These plots are illustrative. However, it is also interesting to have a comparison with other methods used in Federated Learning. It is not clear why n \in {10, 20, 40, 80}  are used and why only m = n/2 is considered.

Additionally, it might be interesting to apply analysis of client sampling to other methods for Federated Learning such as SCAFFOLD (https://arxiv.org/pdf/1910.06378.pdf), Federated Random Reshuffling (https://arxiv.org/pdf/2102.06704.pdf), Variance Reduced methods (http://proceedings.mlr.press/v130/gorbunov21a/gorbunov21a.pdf), but not only localSGD.




**Summary Of The Paper:**

This paper analyses two client sampling strategies for federated learning algorithms. This work compares Multinomial distribution and Uniform distribution. The authors provide a decomposition theorem, that gives some insights into the impact of client sampling. Moreover, they provide convergence guarantees for the general non-convex case under some additional assumptions such as Bounded Dissimilarity. An experimental comparison of two sampling schemes is done at the end of the paper.

**Summary Of The Review:**

Overall, this paper introduces interesting ideas, but it needs significant revision. The assumptions are limited and analysis can be significantly generalized. In this paper, there are no theoretical and experimental comparisons with other methods. Moreover, only two distributions are compared, additional comparison with other distributions is needed.

---

> ### Author Response · Authors · 2021-11-19
> **Authors' Response**
>
> ## Add pseudocode
>
> We have added a pseudocode for FL with client sampling which can be found in Appendix B.
>
>
> ## Assumptions
>
> The reviewer is questioning the basic assumptions behind our theory, especially Assumptions 1 and 3, which are deemed  "questionable", "limited", and non general enough.
> - The reviewer's criticism are motivated based on two works available in arxiv, which are not peer-reviewed (to the best of our knowledge). Could the reviewer provide more compelling arguments to support the claims? We invite Reviewer Sg3m to provide us with peer-reviewed work proving FL convergence with an identical or a faster convergence rate than ours, and with weaker assumptions for nonconvex local loss functions.
> - Most importantly, these assumptions are the building blocks of the ensemble of state-of-the-art works on FL convergence (Wang et al., 2020a; Li et al., 2020a; Karimireddy et al., 2020; Haddadpour &Mahdavi, 2019; Wang et al., 2019a;b). Is the reviewer questioning the domain of FL in general, or is there an argument specific to our work?
>
>
>
> ### Specific answers
>
> **Assumption 1** Khaled et al. (2020) *Better Theory for SGD in the Nonconvex World* proposes a specific modeling of the gradient estiamtor variance to improve Assumption 1. We note that Theorem 2 relies on Assumption 1 for Lemma 3, Lemma 4, and equation (115). By slightly changing the calculus in these places, Assumption 1 can successfully be replaced by the assumption of Khaled et al. (2020). Still, using Assumption 1 makes more sense considering the current state-of-the-art.
>
>
> **Assumption 3**. In Haddapour et al. (2019) *On the Convergence of Local Descent Methods*, authors replace Assumption 1 and 3 by assuming that clients local loss function are $\mu$-Polyak-Lojasiewicz. This assumption is not standard. We note that a $\mu$ strongly convex function is $\mu$-Polyak-Lojasiewicz.
>
> ## Decomposition Theorem 1
>
> For clarity, we replaced "after $K$ SGD on model $\theta^t$" by "after $K$ SGD steps initialized on model $\theta^t$".
>
> In Decomposition Theorem 1, we consider the effect of a single FL step when the clients perform $K$ SGD before the server creates the new global from the participating clients contribution. A variant of Decomposition Theorem 1 can be proposed that holds for any local work method. In equation (79), we replace $\theta_i^{t+1} - \theta^t$ by the stochastic contribution of a client after $K$ SGD step. Decomposition Theorem 1 can instead account for Proximal SGD by repalcing $\theta_i^{t+1} - \theta^t$ accordingly. Still, we see with equation (79) that the influence of $\alpha$ and $\gamma_i$ persists in that case.
>
>
> ## Theorem 2
>
> Theorem 2 is an extension of the theoretical convergence work done in Wang et al. (2020). We insist that with full client participation, Theorem 2 reduces to Theorem 1 in Wang et al. (2020). The authors provide convergence result in nonconvex cases reaching state-of-the-art guarantees. Therefore, our work also does. Providing extensive comparison would be redundant with existing work. We discuss this point in the last paragraph of Section 3.2.
>
> ## Experiments
>
> We consider experimental settings where the influence of a client sampling is exacerbated. Hence, we consider varying number of clients between 10 and 80 with $m = n/ 2$.  With more clients and lower percentage of selected clients, client sampling behavior are closer. We note that reviewer LQPA asked additional experiments where 5-10% of clients are sampled, which we provide in Figure 6 (Appendix E) and confirm our conclusions. Also, we provide experiments where clients perform $K=1$ in Figure 8 (Appendix E) instead of $K=50$ in Figure 2.
>
> Considering the impact of client sampling on other methods would indeed be interesting for future extensions of this work. In our work, we already provide significant results for FedAvg which can be easily extended to a wide range of FL applications with regularization methods, other SGD solvers, and gradient compression/quantization, thanks to the flexibility of the convergence framework.

---

> > ### Comment · Reviewer_Sg3m · 2021-11-22
> > **Response to Author Rebuttal**
> >
> > Thank the authors for their response.
> >
> > I did not find any rules that do not allow me to mention papers from arxiv. The only thing I found is the following:
> >
> > Q: Are authors expected to cite and compare with very recent work? What about non peer-reviewed (e.g., ArXiv) papers?
> >
> > A: ...
> > Authors are encouraged to cite and discuss all relevant papers, but they may be excused for not knowing about papers not published in peer-reviewed conference proceedings or journals.
> >
> > I can understand why authors may not know papers from arxiv, but the argument, that the paper Khaled et al. (2020) Better Theory for SGD in the Nonconvex World is not peer-reviewed and that is why we should not discuss it, seems like an appeal to authority. If the particular paper is not published it does not mean that this paper is not correct or not interesting. Moreover, I did not ask to read the full paper, but only Proposition 1. This paper has many citations, so it means that this paper is quite influential. That is why I wanted to discuss it.
> >
> > I can also mention the paper http://proceedings.mlr.press/v139/gorbunov21a/gorbunov21a.pdf which is accepted to ICML 2021. This paper does not require the bounded variance assumption.
> >
> > The meaning of the Decomposition theorem is still not clear. Moreover, it seems too strict to require other algorithms to satisfy this condition. Future extensions such as compression or random reshuffling might be hard to develop because of the monotonic nature of the proposed framework.
> >
> > Considering this discussion and discussions with other reviewers I believe that my initial score, which is quite high, is appropriate and I would like to keep my score unchanged.

---

> > > ### Author Response · Authors · 2021-11-22
> > > **Authors' Response**
> > >
> > > We don't believe this is the right place for a debate on the value of non-peer reviewed work available in arXiv.
> > >
> > > Our question to the reviewer was precise and addressed the critics made on the FL assumptions used in our work, which are at the basis of the state-of-the-art in FL. Unfortunately we did not receive an answer on this matter.
> > >
> > >
> > > Instead, Reviewer Sg3m cites another paper  Gorbunov et al. *MARINA: Faster Non-Convex Distributed Learning with Compression*. The difference between our assumptions and the ones used in that work are substantial.
> > > In Gorbunov et al. (2021), the authors do not get rid of Assumption 1 and 3 on respectively SGD and bounded dissimilarty, and they instead assume Assumption 3.1. This assumption, which bounds an SGD step in function of the distance between two consecutive global models, is very specific of that work.
> > > Moreover, that work only considers that clients perform a single SGD, i.e. $K=1$, which is not the FL setting. We insist that the assumptions made in our work are standard in FL and cite 6 state-of-the-art papers in federated optimization based on them.

---

### Official Review · Reviewer_mcBg · 2021-11-02

**Correctness:** 3
**Technical Novelty And Significance:** 3
**Empirical Novelty And Significance:** 3
**Recommendation:** 5
**Confidence:** 3

**Main Review:**

1. The main theorem 1 (decomposition theorem) is somewhat too microscopic (as it only studies one round) and more like a direct corollary of the equation (3), and is independent with the other procedure in FL (e.g., local optimization steps). Hence, I would recommend calling theorem 1 "decomposition theorem/lemma for client sampling", rather than the "decomposition theorem for the convergence of FL"

2. The decomposition in theorem 1 is somewhat incomplete as it involves $\theta_i^{t+1}$ outside $Q(\theta^t)$. This makes it hard to interpret the trade-off of the three terms in $Q(\theta^t)$.

3. The paper argues that MD sampling should be used as default case, while uniform sampling is superior only in the special case when clients have the same amount of data. I cannot find a theoretical justification of the first part of this sentence. It is only shown in corollary 1 that Uniform is better than MD in certain cases, but not the reverse side. From Theorem 1, it is clear to me how to argue when MD can be better than Uniform Sampling, as the third term in $Q(\theta^t)$ is always larger in MD.
4. The convergence theorem 2 only studies the regime with sufficiently small local ste
p size $\eta_l$, which effectively reduces to the mini-batch regime. Is it possible to establish  the convergence without assuming sufficiently small local step size $\eta_l$?

Minor comment and suggestions:

1. "The decomposition theorem also provides a necessary condition for an optimization step to improve the current global model: the expected client contribution needs to be collinear with the global direction of the optimum". This claim does not sound accurate. Collinear is neither sufficient or necessary for making progress. The only necessary condition we can read from (7) is the negative inner product. Also, an optimization algorithm does not need to always make progress in distance to optimum every step to "improve" the model.
2. Table 1: the notation $\alpha$ appears much earlier than it was first referenced in the main text (before §3). For better reading experience, it would be great if the authors can hint the semantic of $\alpha$ in the caption of tale 1.

**Summary Of The Paper:**

This work studies the effect of client sampling in the convergence of Federated Learning. This work clearly formulates the setup of client sampling and studies the effect of client sampling to FL progress.

**Summary Of The Review:**

Overall I appreciate author's effort in formulating and understanding the nuances in client sampling. My main concern of this work lies in the significance of the results as well as the interpretation. As this time I think the paper is marginally below the acceptance threshold, but I am happy to re-evaluate the paper if the authors can address my concerns.

---

> ### Author Response · Authors · 2021-11-19
> **Authors' Response**
>
> ## (1. and 2.) Decomposition Theorem 1
>
> We firt recall that the scope of Theorem 1 is not to prove convergence, but instead to provide a clear identification of the impact of client sampling on a single optimization round. This contribution is substantial: the theory of previous state-of-the-art works quantifies the impact of client sampling through (more or less tight) upper bounds which cumulate over iterations, while our equality provides an exact quantification based on interpretable and meaningful statistical quantities.
>
> In particular, the term $Q(\theta^t)$ of Theorem 1 isolates the components impacted by client sampling variance and covariance statistics:  $\nu_i$, $\gamma_i$, and $\alpha$. These quantities are novel in the literature, and allow to draw meaningful conclusions about the role of client sampling, thus motivating the conclusions of the paper. We also note that the analysis of these quantities is independent from $\theta_i^{t+1}$.
>
>
> ## (3.) MD and Uniform Sampling Comparison
>
> The purpose of Theorem is to give insights on a single convergence round, while Theorem 2 provides FL convergence rate for any unbiased client sampling. We introduce two quantities in Theorem 2 :  $\Sigma$ and $\gamma$. Comparison of these two quantities for MD and Uniform sampling tells in which scenarios Uniform sampling is better than MD sampling: in our work we provide sufficient conditions where MD sampling has better convergence guarantees than Uniform sampling. In our work, we only consider the first case, as in practice we have $p_i = \mathcal{O}(1/n)$.
>
> Finally, thanks to our theory we can show that $Q(\theta^t)$ is not always smaller for MD sampling. Indeed, the first term is impacted by $\alpha$ which also impacts the second term through $\gamma_i$. Also, we show with Property 1 that clients aggregation weight variance and covariance are related. As such playing with $\alpha$ cannot be done without changing $\nu_i$ and $\gamma_i$.
>
> Many of these insights have been shown in other works only experimentally, and in this paper we provide for the first time a formal proof of these conclusions (see reviewer LQPA, strength 3).
>
>
> ## (4.) Local Learning Rate $\eta_l$ Conditions
>
> By "small local step size", we assume a bound on the local learning rate $\eta_l$ which we do not put in the statements of Theorem 2 for clarity concerns. Yet, a summary of them can be found in Appendix C.4. As such, we do not assume mini-batch regime as stated by Reviewer mcBg.
>
>
> ## (1.) Improvements in distance
>
> Could Reviewer mcBg be more clear on the concerns about our statement? We are confident on the correctness of Decomposition Theorem 1. As $Q(\theta^t) \ge 0$, a decrease in direction does require the expected direction to be collinear with $\theta^* - \theta^t$.
> Finally, we kindly ask what "improvement" of the model is intended by the reviewer. The distance to the optimum is a quantitative measure of convergence classically adopted in converence analysis. As such, the equality of Theorem 1 is informative of the convergence process and relevant to the scope of this work.

---

> > ### Comment · Reviewer_mcBg · 2021-11-21
> > **Response to Author Rebuttal**
> >
> > I thank the authors for the detailed response.
> >
> > > Could Reviewer mcBg be more clear on the concerns about our statement?
> >
> > I agree the decomposition theorem is technically correct, but I am concerned with the interpretation of this theorem.
> >
> > 1. "collinear" is not the correct term here -- collinear typically means two vectors are along the exact same direction, but here it seems more like cosine > 0.
> > 2. An optimization algorithm does not have to be monotonic in $\|\theta^* - \theta^t\|$ every step to converge. (For example, Nesterov accelerated GD does not satisfy this guarantee, but converges faster w.r.t. an alternative potential). Therefore it is not accurate to claim that the decomposition theorem is a necessary condition for an algorithm to converge.
> >
> > > As such, we do not assume mini-batch regime as stated by Reviewer mcBg.
> > My claim was not the paper studied mini-batch per se, but instead by setting $\eta_l$ sufficiently small and server $\eta$ sufficiently large, it effectively reduces to the mini-batch convergence regime.
> >
> > ---
> > This microscopic issue is my main concern in this work. A monotonic condition on a particular potential of a single step has limited guiding significance of the overall algorithm.
> >
> > After reading the response and reviews of the other reviewers, I have decided to keep my original evaluation.

---

> > > ### Author Response · Authors · 2021-11-22
> > > **Authors' Response**
> > >
> > > As stated by Reviewer mcBg, the purpose of Decomposition Theorem 1 is indeed to give insights on the impact of client sampling on the resulting global model $\theta^{t+1}$ distance with respect to the optimum $\theta^*$. However we remind Reviewer mcBg that our equality is about a single optimization round, and not the whole optimization process.
> > >
> > > We still do not find any contradiction in our statement "The Decomposition Theorem provides a necessary condition for an optimziation step to improve the current global model". The client drift needs to be negative for the global model to get closer to the optimum.
> > >
> > > Concerning the statement "A monotonic condition on a particular potential of a single step has limited guiding significance of the overall algorithm". We note that the insights provided by Decomposition Theorem 1 are proven in Theorem 2 for the whole learning process. As such, we kindly ask Reviewer mcBg to explain in what the convergence bound of Theorem 2 does not explain the impact of a client sampling on FL convergence.

---

### Official Review · Reviewer_FJmg · 2021-11-02

**Correctness:** 4
**Technical Novelty And Significance:** 2
**Empirical Novelty And Significance:** 2
**Recommendation:** 3
**Confidence:** 3

**Main Review:**

**Pros**:
- The paper is well-written
- The theory is well-developed and is correct to my knowledge

**Cons**:
My major concern is on the novelty of the theoretical analysis of clients' sampling.

- Clients sampling in FL and mini-batch sampling in SGD are closely related to each other. Related works on the sampling strategy for SGD are not well-address, for example [1, 2]. The major difference between FL and mini-batch SGD is that FL runs more steps during the local update, but SGD performs a single step. But this difference should not cause too much trouble. It seems that the analysis of the sampling scheme for SGD can be directly applied to the context of FL. Could the author(s) elaborate on the challenge of applying importance sampling to FL?

- Unbiased client sampling is closely related to unbiased gradient estimator, which is common knowledge in stochastic optimization. The discussion on unbiased sampling doesn't
seem to be something new.

- The authors emphasized the importance and novelty of Theorem 1. I read the proof, and it seems to me that Theorem 1 is a detailed expansion of the variance term
(instead of just bounding the variance by some constant in the literature). It is also straightforward that the variance can be decomposed as the summation of covariance.
The Theorem feels somewhat incremental, and I am not sure if it is significant enough to be accepted in ICLR.

- My comment on Theorem 2 is similar to the above comment for Theorem 1. The result is from a detailed expansion of the variance term, and I am not sure if it is significant enough.

Some minor comments:
- In eq. (1), is it necessary to introduce "I"? Can we just write as \sum_{i=1}^n?
- In the second paragraph of page 7. The author(s) said that MD has an additional O(n) cost to construct the distribution and could be expensive. I think we only need to run this operation once before the FL start. I doubt if constructing the probability density for MD will introduce non-trivial overhead.


[1] Peilin Zhao, Tong Zhang. Stochastic Optimization with Importance Sampling for Regularized Loss Minimization. ICML 2015.

[2] Dominik Csiba, Peter Richt´arik. Importance Sampling for Minibatches. JMLR 2018.


**Summary Of The Paper:**

The author(s) studied the impact of the client sampling for FL. In particular, the author(s) proposed a decomposition theorem and used it to obtain an improved convergence analysis of FedAvg. Numerical simulations are conducted to support the author(s) theoretical analysis.

**Summary Of The Review:**

The paper is well-organized. But I am not convinced by the significance of the theoretical analysis of this paper.

---

> ### Author Response · Authors · 2021-11-19
> **Authors' Response**
>
> ## Client Sampling and Mini-batch Sampling
>
> The reviewer points to the analogy between FL and mini-batch SGD: with client sampling in FL, the server selects a subset of clients to ask partiticipation from, while with mini-batch SGD, the server selects sample to perform one SGD step. We respectfully disagree with the reviewer on the equivalence between these problems in general, since optimization in FL is not straightforward and presents unique challenges. For example, both works [1] and [2] pointed by Reviewer FJmg propose mini-batch sampling where samples importance is given according to specific data information which is not available in FL (knowledge of the Lipschitz smoothness $L_i$ of a sample loss function [1], or the samples knowledge parameteres $v_i$ [2]). These parameters are data dependent and thus private  in FL. In this case, the proposed mini-batch SGD theory cannot be applied. This justifies the interest of data agnostic approaches including MD sampling and Uniform sampling.
>
> ## Unbiased Client Sampling
>
> We do not discuss in this work the relevance of unbiased client sampling, and we agree that "unbiased client sampling is closely related to unbiased gradient estimator", as shown in (Wang et al., 2020, and Li et al. , 2020). The contribution of our work is in showing the impact of client sampling variance (and covariance) on FL convergence speed.
>
> ## Decomposition Theorem 1 and Theorem 2
>
> Decomposition Theorem 1 highlights the impact of client sampling on a single convergence round. The strength of this theorem lies in relating with an equality two consecutive global model distances with the optimum, and not in proving FL convergence. While the derivation is deemed as straightforward by Reviewer FJmg, no previous work has proposed this decomposition before. More precisely, no one ever showed the role of the sampling covariance on FL convergence. This contribution is substantial: the theory of previous state-of-the-art works quantifies the impact of client sampling through (more or less tight) upper bounds which cumulate over iterations, while our equality provides an exact quantification based on interpretable and meaningful statistical quantities.
>
>
> ## Time complexity to sample clients
> For MD sampling, $\mathcal{O}(n)$ comes from the construction of the distribution and should not change during the learning process. This operation needs to be only run once at the beginning of the learning process. Still, once the probability density function is built, Uniform sampling requires time complexity $\mathcal{O} \left(m\log (n/m) \right)$ to sample $m$ clients while MD sampling requires $\mathcal{O} \left(m \log(n)\right)$.

---

> > ### Comment · Reviewer_FJmg · 2021-11-19
> > **Thanks for your respond**
> >
> > I would like to thank the author(s) for the clarification. I agree that the importance sampling used in MB-SGD would require additional information from local data to do the sampling and therefore can not be trivially applied to federated learning, I think this message is not trivial and may be worth using one short paragraph to discuss why importance sampling is not applicable.
> >
> > For Theorem 1 and Theorem 2, I am still not fully convinced by its significance. Although it has not appeared explicitly in the literature, its technical contribution still seems incremental to me, which is the main concern I have for this paper. Overall I would like to keep my score unchanged.

---

> > > ### Author Response · Authors · 2021-11-22
> > > **Authors' Response**
> > >
> > > We thank Reviewer FJmg for acknowledging that mini-batch sampling and client sampling for Federated Learning are two different research topics. In our rebuttal we have answered every concern Reviewer FJmg had, and the reviewer seems to acknowledge the novelty of our theorem.
> > >
> > > Nevertheless, our work is still found "incremental" without precise elements on the incremental nature of our contribution.

---

### Official Review · Reviewer_LQPA · 2021-11-02

**Correctness:** 4
**Technical Novelty And Significance:** 3
**Empirical Novelty And Significance:** 2
**Recommendation:** 6
**Confidence:** 5

**Main Review:**

Strengths:
- The paper is very well written and very easy to understand and follow.
- The paper studies a framework for studying the impact of client selection strategies for unbiased aggregation at the server. The framework separates the impact of client selection, client model drift, global model drift thereby providing new insights regarding the impact of client selection.
- Through the framework, the authors provide theoretical and empirical evidence as to in which scenarios weighted sampling is preferrable over weighted sampling. Previously multiple papers had observed stable  convergence performance  with weighted sampling for cases  data sample is weighed equally. This framework validates those claims.
- The experimental results back the theory.

Weaknesses:
- As showcased in Wang, et.al., "Tackling objective inconsistency in federated learning" (FedNova), the convergence rate in FL is affected by the objective inconsistency as well. It is not clear from the decomposition theorem, whether such an inconsistency is factored in.
- It is not clear from the experiments what sort of data distribution was considered for the clients. It is not clear if the claims made in the paper continue to hold for both iid and non-iid distributions.
- Moreover, the number of clients selected is roughly 50% of all clients present in the experiments. With unbiased aggregation, the variance is controlled with such a high selection rate. It is not clear if the number of clients are selected at a rate 5-10% clients per round or even lower, which is characterized by zero bias and high variance would have a different observation. With variance being the main deciding factor now, is there still a noticeable difference between sampling strategies? Also, how noticeable is the difference when each client undergoes just one step of SGD or likes of that as opposed to 50 steps of SGD. I would recommend the authors to add more experiments with only 5-10% clients being selected in each round.

Finally, as a suggestion to the authors: In most practical systems, the server doesn't have the privilege to sample clients based on MD or uniform sampling. But, instead the sampling in practical FL system is mostly event triggered. In such a case, how would one go about sampling clients and ensure unbiasedness of aggregation schemes?

**Summary Of The Paper:**

The paper studies a framework for studying the impact of client selection strategies for unbiased aggregation at the server for federated learning. The framework separates the impact of client selection, client model drift, global model drift thereby providing new insights regarding the impact of client selection. In particular, the paper proposed a decomposition theorem which quantifies  the dependence on client sampling. Finally, experimental results on Shakespeare back the theory and demonstrate the effectiveness of one sampling choice over another based on the loss function setup in a FL framework.

**Summary Of The Review:**

This paper proposes a new framework to study the effect of client sampling strategies based on weighting of clients/samples in a federating learning setup. While the framework is new and provides new insights, there are some clarifications required in terms of the theoretical convergence results. The experimental results are also incomplete and not up to the mark. Additional experiments are required. The paper is in a good shape, but without the improvements above it's not a strong paper yet.

---

> ### Author Response · Authors · 2021-11-19
> **Authors' Response**
>
> ## Heterogeneous Amount of Local Work
>
> The reviewer is right when stating that Wang et al. (2020) show that FedAvg can lead to minimization of a surrogate objective to the original one, equation (1), for example when clients perform heterogeneous amount of local work. We emphasize that our Decomposition Theorem 1 is not a substitute to FL convergence guarantees, and its purpose is to provide insights on the impact of client sampling on a single optimization round.
>
> This being said, the influence of the objective inconsistency can be appreciated in the direction drift term of the proposed decomposition. Indeed, in this case $p_i$ is replaced by the importance $w_i$ of client $i$ in the surrogate ojective, which makes more challenging for the expected client contribution to be collinear with $\theta^* - \theta^t$.
>
>
> ## Experiments
>
> Figure 2 is obtained using the federated version of Shakespeare dataset. Clients receive all the sentences of one play character in a Shakespeare play for a next-character prediction task. More details on the construction of the dataset can be found in Caldas et al. (2018). The authors made their code publicly available. By construction, a client data distribution is unique and thus non-iid.
>
> To show the influence of client sampling on FL convergence rate, we consider a learning application where data distributions vary significantly between clients (Shakespeare), with few clients ($n \in [10, 20, 40, 80]$). As a result, the server has interest in sampling as many different clients as possible to maximize the data support at the server level. To maximize the difference in the client sampling metrics, we also consider $m = 0.5 n$.
>
> Our theoretical and experimental conclusions remain true but less noticeable when less clients are participating. We provide in Appendix E Figure 6 these results, where we consider $n=80$ with $m =0.05\times n = 4$ and $m =0.1\times n = 8$.
>
>
> We also show the impact of client sampling on the convergence is proportional to the amount of local work, but cannot be mitigated by considering K=1 (Figure 8 in Appendix E).
>
>
> ## Event triggered-aggregations
>
> We consider in our work that at every optimization round, the server selects a subset of clients based on their data ratio $p_i$ and a client sampling strategy. This scenario is standard and used to reduce clients computation workload and improve the speed of the optimization rounds. As a result, in this case, the server knows the aggregation weight given to the clients before asking them to work.
>
> The reviewer points to the setting where "the server doesn't have the privilege to sample clients" and aggregation is "event-triggered". This setting is more challenging since the server needs to know every client participation probability to properly allocate each client a weight such that  $\mathbb{E}\left[\omega_i(S_t)\right] = p_i$ is satisfied. In practice, knowing this probability is not possible but the server can assume that clients participate according to a given distribution. For example, Li et al. (2020) assume that clients uniformly update their work to the server. Even in this setting, while alternative schemes can be considered, our theory would still holds and provides a quantification of the impact on FL convergence.

---

> > ### Comment · Reviewer_LQPA · 2021-11-30
> > **Thanks for the response!**
> >
> > Thanks to the authors for the detailed response! The paper is a solid theoretical contribution but it can be improved in terms of experimentation. I would like to maintain my score.

---

> ### Comment · Reviewer_c4xq · 2021-11-20
> **Insights**
>
> Dear reviewer LQPA,
> Since the authors quote your review, in particular strength 2, in their response to my review, I would like to ask if you could elaborate on the insights of this work that you mention a few times in your review. Do you think it can somehow help with the selection of sampling strategies or provide intuition for practical implementation? I would particularly appreciate it if you could point to a specific result or corollary that you find useful.

---

> > ### Comment · Reviewer_LQPA · 2021-11-30
> > **Response**
> >
> > Dear reviewer c4xq,
> >
> > All FL algorithms proposed till date provide their convergence results agnostic to client sampling. Most convergence results are stated only in terms of client and global drift. The insights provided in this paper explicitly factors out the client sampling dependence. The authors don't exhaustively search for the best sampling strategy out there, but I believe and hope this result would lead to more carefully tuned device selection policies as opposed to standard/weighted sampling.

---

### Official Review · Reviewer_c4xq · 2021-11-04

**Correctness:** 4
**Technical Novelty And Significance:** 2
**Empirical Novelty And Significance:** 2
**Recommendation:** 3
**Confidence:** 4

**Main Review:**

## Main point:
I don't see a sufficient contribution in this work for its acceptance. First of all, most of the results are straightforward. The results from Appendix A require no more than computing expectations and simple summations. Theorem 1 is not even a convergence result but rather a simple one-step recursion that can be obtained by expanding norms and scalar products. Theorem 2 is proved using standard techniques from prior work. Taking all of this into account, I feel like the technical contribution is extremely limited.

In addition, I do not see any significant insight into the convergence of FedAvg. The impact of client sampling has been studied before and optimal sampling strategies were proposed by Chen et al. (2020), and other works, for instance (Fraboni et al., 2021), have studied more forms of unbiased sampling. This work gives a bit more detail into how it all affects training but it all seems trivial, but without new improved strategies for sampling, it is hard to see the value of this study.

### Minor concerns:
I think it's a bit confusing to denote the constant in Assumption 3 as $\kappa$ since $\kappa$ is often used to denote the problem conditioning.

### Typos:
"on a fix amount" -> "on a fixed amount"
"dependence of FL convergence from the variance" -> "dependence of FL convergence on the variance"
"through a fixed amount K of SGD initialized" -> "through a fixed amount K of SGD steps initialized"
"needs to be collinear with the global direction" -> "needs to be correlated with the global direction"
"Our work is structured as follow." -> "Our work is structured as follows."
Assumption 3: "There exists constants" -> "There exist constants"
The right-hand side of Equation (63) misses square brackets after "Cov".

**Summary Of The Paper:**

This paper considers the problem of Federated Learning (FL) with a random selection of clients in order to minimize $\sum_{i} p_i \mathcal{L}_i(\theta)$, where $\mathcal{L}_i$ is the loss function of client $i$ and $p_i$ is its weight, which is often proportional to the dataset size on client $i$. The authors study the impact of assigning weights $\omega_i(S_t)$ and probabilities $P(i\in S_t)$ to client $i$ when averaging the result of running SGD for $K$ steps locally on each sampled device. The authors first prove a decomposition bound (Theorem 1), which establishes a simple recursion for the distance to the solution. As far as I can see, this result is never used to establish a convergence rate. In addition to Theorem 1, under smoothness, bounded variance, and bounded dissimilarity assumptions, the authors study convergence rates of unbiased client sampling in the nonconvex regime (Theorem 2). As a side contribution, the authors also consider using different stepsizes $\eta_l, \eta_g$ locally and globally.

**Summary Of The Review:**

1. I see very little novelty in this work. Most results follow by combining some simple identities for expectations of random variables with convergence results for Local SGD from prior literature (Wang et al., 2020a), (Li et al., 2020c). Adding a global stepsize also does not change almost anything in the proofs.
2. The results do not have a clear significance. While Federated Learning is an important and challenging problem, and client sampling may play a big role in its efficient use in practice, the results in this paper provide us with little new insight.

---

> ### Author Response · Authors · 2021-11-19
> **Authors' Response**
>
> ## Contribution and Previous Work
>
> Unfortunately we do not see the rationale for dismissing our contribution because our theory is not perceived as difficult enough. The main contribution of this work is Theorem 2 providing convergence guarantees for any unbiased client sampling. Our mathematical derivation is correct, and although we may rely on classical tools from statistical calculus, this is the standard for most of several state-of-the-art works in FL. Could the reviewer be more specific about what is trivial in the 5 pages of derivations provided in Appendix C?
>
> On a more fundamental aspect, we acknowledge that our works extends the framework of Wang et al. (2020), and we emphasize that our contribution is indeed the extension of that theory, to account for any client sampling strategy (aspect which is missing in Wang et al. (2020)). The novelty and strength of this contribution has been recognized also by reviewer LQPA (strengths 2 and 3).
>
> Our work provides convergence guarantees for FedAvg in function of a client sampling statistics. Besides the aggregation weight variance, this is the first work demonstrating that the sampling covariance plays also an important role in convergence. This aspect is novel and is the first one of this kind.
>
> Considering the papers mentioned by the reviewer:
>
> **Chen et al. (2020)** introduces *optimal sampling*, a new client sampling where clients perform their local work and send its norm to the server. Based on that norm, the server computes a probability $q_i$ and clients send their contribution with a Bernoulli $\mathbb{I}(q_i)$.
> This sampling strategy is not among the standard sampling strategy of FL, while our theory applies to any strategy, especially the ones which are the state-of-the-art in FL.
>
> In this sense, their work does not provide any theoretical guarantees regarding FL convergence for any other client sampling including MD sampling and Uniform sampling. In our work, we bridge this gap by providing a general framework that can account for a wide-range of client sampling (see Appendix A). In addition, we added in Section A.8 the close form of the aggregation weight variance and covariance parameter $\alpha$ proposed in Chen et al, and as such our work still holds for their strategy. This is another proof of the generality and relevance of our theory.
>
> **Fraboni et al. (2021)** introduces *clustered sampling* to improve MD sampling. Instead of sampling $m$ clients from the same distribution, the server creates $m$ distributions and sample a client from each of them. As such, MD sampling is a special case of clustered sampling where all the distributions are identical.
>
> In Fraboni et al. (2021), the authors only provide (using also the theoretical framework of Wang et al. (2020)) identical convergence guarantees for both MD and clustered sampling while experimentally proving that clustered sampling outperforms MD sampling. In our work, we provide more general convergence guarantees enabling comparison of the convergence for both MD and clustered sampling (see Appendix A.6 and C.5)
>
> Reviewer c4xq states "the results in this paper provide us with little new insight.". We fail to retrieve in Chen et al. (2020) and Fraboni et al. (2021) the theoretical conclusions we show in our work. Could the reviewer please indicate us where to find such conclusions?
>
> ## Decomposition Theorem 1
>
> While the current literature focuses on providing convergence guarantees, the Decomposition Theorem 1 provides for the first time an equality to quantify the impact of client sampling on each optimization round. This contribution is substantial: the theory of previous state-of-the-art works quantifies the impact of client sampling through (more or less tight) upper bounds which cumulate over iterations, while our equality provides an exact quantification.

---

> > ### Comment · Reviewer_c4xq · 2021-11-20
> > **Clarifications**
> >
> > I thank the authors for posting their comments. I address them with quotes below.
> > > Could the reviewer be more specific about what is trivial in the 5 pages of derivations provided in Appendix C?
> >
> > The results in Appendix C follow almost the same lines as the proofs of (Wang et al., 2020a). Your equation (89) gives us the update in the form of a weighted average and is the same as equation (46) in (Wang et al., 2020a), and equation (91) computes the expectation of (89), essentially replacing random weights with their expectations. Then we have Lemma 3, whose proof is a modification of the derivation by Weng et al. (2020a). Your decompositions into $T_1$ and $T_2$ are the same, up to constant factors and weights, as in (Wang et al., 2020a). Your Lemma 4 follows the same steps as the derivations in Appendix C.5 of (Wang et al., 2020a), namely from their equation (73) to (87). It seems to me that you tried to hide this by doing some minor changes, for instance slightly changing the definitions of $T_1$ and $T_2$ or replacing constant $D$ of (Wang et al., 2020a) with your own constant $R$. One change is in the upper bound on $T_2$ but the derivation itself is the same.
> > I also would like to note that introducing server stepsizes is an effortless modification since it does not require any additional steps in the proofs.
> > >The novelty and strength of this contribution has been recognized also by reviewer LQPA (strengths 2 and 3).
> >
> > As I mentioned in my review, "I do not see any significant insight into the convergence of FedAvg". In contrast, Reviewer LQPA refers to new insights in their strength 2. I will Reviewer LQPA for specifics about this.
> > >this is the first work demonstrating that the sampling covariance plays also an important role in convergence. This aspect is novel and is the first one of this kind.
> >
> > I think "novel" and "the first one of this kind" mean mostly the same thing. However, what I fail to see is why results of this kind are desired in the first place.
> > >We fail to retrieve in Chen et al. (2020) and Fraboni et al. (2021) the theoretical conclusions we show in our work. Could the reviewer please indicate us where to find such conclusions?
> >
> > Of course, the works of Chen et al. (2020) and Fraboni et al. (2021) do not answer all questions in federated learning, and their works do not cover every statement presented in this submission. The point I am trying to make is that I fail to see why the motivation behind deriving the statements in this submission. The motivation of Chen et al. (2020) is clearly to get optimal sampling that would perform well in practice. Since we can hope to see faster convergence, I understand that the work of Chen et al. (2020) may have some value. What I do not see is the value of the results in this submission.

---

> > > ### Author Response · Authors · 2021-11-22
> > > **Authors' Response**
> > >
> > > The reviewer makes the following statement:
> > >
> > > "It seems to me that you tried to hide this ...".
> > >
> > > We kindly ask to the reviewer to revise the tone of this statement, and avoid making this kind of insinuations on our intentions.
> > >
> > > ## Proof of Theorem 2 with respect to the work Wang et al., 2020a
> > >
> > > Is is clearly stated in our derivation that our work is an extension of Wang et al. (2020) (e.g. Appendix C, first line). However, starting from Wang et al. (2020), we introduce quantities and derivations which are novel, and at the core of our contribution.
> > > More specifically:
> > >
> > > - **Regarding $T_1$ and $T_2$**. We keep the same denomination as Wang et al. (2020) thus denoting $T_1$ and $T_2$ the two elements in equation (106). $T_1$ is developepd in two lines identically to Wang et al. (2020) as client sampling does not change anything. However, the derivation for $T_2$  is different from Wang et al. (2020), cf equation (112) to (115), which introduces a quantity we bound with **Lemma 4 not included** in Wang et al. (2020).
> > >
> > > - **Regarding $D$**. Indeed, we rename the constant $D$ in Wang et al. (2020) by $R$ in our work. This decision is easily justified by the final bound that we obtain. The one in Wang et al. (2020) depends of three terms denoted from $A$ to $C$, while ours depend of four terms and we thus need letter $D$ for our convergence bound. Again, we never attempted at claiming the contribution of Wang et al. (2020)  as ours.
> > >
> > > - **Regarding $\eta_g$**. We never state that including a global learning  $\eta_g$ makes the proof of Theorem 2 more challenging. Yet, it is useful to understand the impact of $\eta_g$ on FedAvg convergence bound. Indeed, $\eta_g$ can be used to mitigate clients contribution discrepancy due to their heterogeneous data distributions.
> > >
> > > - **Regarding Lemma 3**. We do not claim to prove this Lemma, and clearly state that "The proof is in Section C.5 of Wang et al. (2020)".
> > >
> > > - **Regarding Lemma 4**. We use the work of Wang et al. (2020) to obtain only equation (99), which we acknowledged with the statement "Section C.5 of Wang et al. (2020) proves ...".
> > >
> > >
> > > ## Novelty and significance
> > >
> > > The reviewer is questioning the "value of the results in this submission" and states that "fails to see is why results of this kind are desired in the first place".
> > >
> > > We reiterate that the main contribution of our work is to be the first one providing convergence guarantees (Theorem 2) depending on client sampling statistics: **variance and covariance**. While several client samplings have been proposed for FL, no theoretical framework enables to compare their behavior on the convergence speed.
> > >
> > > The reviewer still criticizes our work in light of 1) perceived technical complexity (see comments above), and 2) the papers Chen et al. (2020) and Fraboni et al. (2021), which have however different focus from ours. The contribution of this work is to show the impact of a client sampling on FL convergence rate and not to provide a new client sampling.  We reiterate that no theoretical work has yet been proposed to understand the advantage of any general client sampling strategy, which our theory can do.

---

> > > > ### Comment · Reviewer_c4xq · 2021-11-29
> > > > **Sorry for the tone**
> > > >
> > > > > We kindly ask to the reviewer to revise the tone of this statement, and avoid making this kind of insinuations on our intentions.
> > > >
> > > > I apologize for my tone. The change in notation looks strange to me taking into account the similarity of the proofs, but I shouldn't have written that the authors tried to hide it, it is indeed clearly stated that the proof is based on that of Wang et al.
> > > >
> > > > > We never state that including a global learning $\eta_g$ makes the proof of Theorem 2 more challenging
> > > >
> > > > Thanks for this clarification, in that case, I will view this aspect of the theory as secondary.
> > > >
> > > > > We reiterate that the main contribution of our work is to be the first one providing convergence guarantees (Theorem 2) depending on client sampling statistics: variance and covariance.
> > > >
> > > > My issue with this claim is that it is not obvious per se that obtaining this result for the first time is valuable. I could understand the value of the result if there were some consequences of the theory. It appears to me that the only suggested benefit is the generality of the results. However, more general does not necessarily mean better, unless there is clear evidence that we need more general results.
> > > >
> > > > > The reviewer still criticizes our work in light of 1) perceived technical complexity (see comments above), and 2) the papers Chen et al. (2020) and Fraboni et al. (2021), which have however different focus from ours. The contribution of this work is to show the impact of a client sampling on FL convergence rate and not to provide a new client sampling.
> > > >
> > > > I referred to the papers of Chen et al. (2020) and Fraboni et al. (2021) to illustrate my point that I find providing new client sampling strategies a bigger contribution than showing the impact of arbitrary client sampling of the convergence rate.

---

### Author Response · Authors · 2021-11-19
**General Rebuttal**

We thank the reviewers for the their time and for comments given to our paper.
A number of points raised by the reviewers have helped us in improving the manuscript, especially in clarity (Reviewer FJmg and mcBG), experimental setting (Reviewer LQPA and Sg3m), and in the positioning with respect to the state-of-the-art.
Nevertheless, we find the relevance of certain  critical comments debatable:

- The main critic of Reviewer c4xq seems based on the perceived difficulty of the derivations of our theorem. "Expectations and simple summations" are the working tools of statistical calculus, used in most of state-of-the-art works in FL. We would appreciate if the reviewer could be more specific on what is found trivial in our  derivations.
- Reviewer FJmg is questioning the relevance and originality of FL optimization/sampling with respect to the field of mini-batch sampling in SGD. The reviewer seems to miss the fundamental motivation of FL optimization (non-accessibility of local data information) which is specific of FL convergence theory.
- Reviewer Sg3m is questioning the basic assumptions of the state-of-the-art theory on FL convergence. These concerns are motivated by non-peer reviewed material, and are not specific to our contribution.

We hope that our point-to-point responses can clarify the concerns expressed by the reviewers, and better highlight the contribution of our study.

---

### Decision · Program_Chairs · 2022-01-20

**Decision:**

Reject

**Comment:**

Dear authors,

I have carefully read the reviews, rebuttals, and the subsequent discussion. Most reviews provide high quality feedback, and the reviewers' combined opinion is strongly oriented towards recommending rejection. I have to concur with this recommendation. While essentially all reviewers agreed that the paper is well written, they also raised several key concerns. I will reiterate (and further elaborate) on some of them here:

1) I do not agree with the authors' claim that there is a significant difference between client sampling in FL and data sampling in SGD in terms of the underlying mathematics; at least not in the present form. The mathematical formulation of both problems is the same. While in SGD it is possible to access local information and construct more powerful sampling strategies using this information, it is also possible to forgo using this information and propose simpler data-agnostic strategies. Such strategies have been studies in the SGD literature before. I recommend the literature on *arbitrary sampling* pioneered by Richtarik and Takac ("Iteration complexity of randomized block-coordinate descent methods for minimizing a composite function", Mathematical Programming, 2014) in the context of randomized coordinate descent. The same sampling approach was later adopted for SGD. The paper by Csiba and Richtarik (Importance sampling for minibatches, JMLR 2018) suggested by one of the reviewers is relevant here as it adopts the arbitrary sampling approach to (variance reduced) SGD. Such an arbitrary sampling paradigm is more general than the unbiased sampling strategy you study here (indeed, arbitrary sampling includes also biased samplings). The work of Chen et al mentioned by a reviewer is also more relevant than you appreciate. This work also mentions a couple decomposition statements. They work with the arbitrary sampling framework in the FL setting - and this seems more general than your framework since it includes biased samplings as well. Note that they then proceed to compute the optimal sampling out of all samplings, while you do not attempt to theoretically capture what samplings are best. This prior works thus addresses a similar problem, and goes deeper in this aspect. Also, the parameters $v_i$ in their Lemma 1 are simply just statistics of the sampling - and are unrelated to the data. Also, Lipschitz smoothness constant of the aggregate loss over all local data samples is not hard to estimate, and is needed anyway to set the stepsize correctly. So, your comments about the unavailability of these quantities in the FL setting seem incorrect.

2) Theorem 1 is indeed just a simple calculation / observation rather than a result. I agree with the reviewer who said that the value of this simple observation, without a deeper study of its consequences, and an explanation of how the consequences lead to new results that are in some sense interesting, is quite limited. As mentioned by one reviewer, sophisticated methods are often non-monotonic: they do *not* attempt to greedily reduce some simple potential (e.g., distance to the minimizer) as such a strategy may be suboptimal from a total convergence point of view. This observation by the reviewer seems to have been misunderstood by the authors. This limits the impact of Theorem 1 as Theorem 1 assumes that one is interested in a greedy method.

3) The bounded variance and bounded dissimilarity assumption *are* strong. The suggestion by one of the reviewers to consider a work on more accurate ways of modeling stochastic gradients (Khaled et al) was appropriate. I suggest the authors read the paper to see detailed reasoning explaining why the types of assumptions you make in this paper are problematic. The fact that some other papers use such problematic assumptions, even if they are well known, is not evidence that these assumptions are not problematic. It is merely evidence that many papers share the same issue. I also have to oppose the authors' view that non-peer-reviewed work should not be brought up. In my view, this is a deeply problematic and unscientific attitude to research that is available online. Peer review does not imply correctness, and vice versa.

4) Experimental comparison with any other methods is missing. Why do you not compare with the optimal sampling strategy of Chen et al, for example? Does your framework suggest a better strategy in some sense? If yes, show it. If no, then in what sense are your sampling strategies interesting? In any case, all have been considered in the arbitrary sampling framework before as far as I can see.

5) There are more issues with have been identified by the reviewers. I strongly recommend the authors to take all of them seriously in their revision.

In summary, this is a solid paper. However, it has some serious issues and for this reason I cannot recommend it for acceptance. Having said that, I thank the authors for their submission and wish them best of luck in future research with this project.

Area Chair